# TRAINING ROBUST ENSEMBLES REQUIRES RETHINKING LIPSCHITZ CONTINUITY

**Ali Ebrahimpour Boroojeny**
UIUC
ae20@illinois.edu

**Hari Sundaram**
UIUC
hs1@illinois.edu

**Varun Chandrasekaran**
UIUC
varunc@illinois.edu

## ABSTRACT

Transferability of adversarial examples is a well-known property that endangers all classification models, even those that are only accessible through black-box queries. Prior work has shown that an ensemble of models is more resilient to transferability: the probability that an adversarial example is effective against most models of the ensemble is low. Thus, most ongoing research focuses on improving ensemble diversity. Another line of prior work has shown that Lipschitz continuity of the models can make models more robust since it limits how a model's output changes with small input perturbations. *In this paper, we study the effect of Lipschitz continuity on transferability rates.* We show that although a lower Lipschitz constant increases the robustness of a single model, it is not as beneficial in training robust ensembles as it increases the transferability rate of adversarial examples across models in the ensemble. Therefore, we introduce LOTOS, a new training paradigm for ensembles, which counteracts this adverse effect. It does so by promoting orthogonality among the top-$k$ sub-spaces of the transformations of the corresponding affine layers of any pair of models in the ensemble. We theoretically show that $k$ does not need to be large for convolutional layers, which makes the computational overhead negligible. Through various experiments, we show LOTOS increases the robust accuracy of ensembles of ResNet-18 models by 6 percentage points (p.p) against black-box attacks on CIFAR-10. It is also capable of combining with the robustness of prior state-of-the-art methods for training robust ensembles to enhance their robust accuracy by 10.7 p.p. The code is publicly available at https://github.com/Ali-E/LOTOS.

## 1 INTRODUCTION

Deep learning models are very brittle to input changes: Szegedy et al. (2013) were the first to carefully craft "adversarial" perturbations that result in incorrect classification outcomes. Subsequent research showed that these adversarial examples "transfer" to models with different hyper-parameters, and even different hypothesis classes (Papernot et al., 2016a; Liu et al., 2016). This transferability property was used to design black-box attacks against models for which only query access is available (Papernot et al., 2017). In these attacks, the adversary trains a local model that is "similar" to the victim (black-box) model and uses that to find transferable adversarial examples (Xiao et al., 2018). To increase the rate of transfer success, a common strategy is to choose those inputs that are adversarial to an "ensemble" of models (Liu et al., 2016; Chen et al., 2023): fooling an ensemble may result in fooling a potentially unseen model.

Research has also been carried out on understanding and improving model resilience to such attacks (Dong et al., 2019). Adversarial robustness is the innate ability of the model to correctly classify adversarial examples (Madry et al., 2017). One way of increasing this robustness is utilizing a diverse[1] ensemble of models (Yang et al., 2020; 2021). This has also been considered as a mitigation to the transferability problem (Pang et al., 2019; Kariyappa & Qureshi, 2019; Yang et al., 2020; 2021; Sitawarin et al., 2023); by increasing the diversity among models of the ensemble, the subspace of the adversarial examples that are effective against most of the models within the ensemble shrinks.

---

[1]In terms of the parameters and decision boundaries.

While empirical evidence is promising, most of these ensemble robustness methods are either computationally expensive or come at a considerable cost to the accuracy of the models and the overall ensemble. In another vein of research, Lipschitz continuity was also shown to be important for robustness (Szegedy et al., 2013; Farnia et al., 2018; Boroojeny et al., 2024; Ebrahimpour-Boroojeny et al., 2025). Since the model's Lipschitz continuity controls how the predictions change for small changes in the input, it is intuitive that bounding it will improve robustness. It is important to note that neural networks are a composition of multiple layers and therefore the existing works in this area obtain an upper-bound on the overall Lipschitz constant of the network by bounding each layer independently (Szegedy et al., 2013; Sedghi et al., 2018; Senderovich et al., 2022; Delattre et al., 2023; Boroojeny et al., 2024).

In our work, *we investigate the effect of Lipschitz continuity on the transferability of adversarial examples*. We observe that while decreasing the Lipschitz constant makes each model of the ensemble *individually more robust*, it makes them less diverse and consequently *increases the transferability rate* among them which in turn *hinders the overall ensemble robustness* (§ 3, Figure 1). To resolve this adverse effect, we introduce our novel training paradigm, **Layer-wise Orthogonalization for Training rObust enSembles (`LOTOS`)**, which orthogonalizes the corresponding affine layers of the models with respect to one another. This increases the diversity of the trained models. `LOTOS` can be combined with any prior method of training diverse ensembles to further boost their robustness.

Through extensive experiments and ablation studies, we show that `LOTOS` effectively decreases the transferability rate among the models of an ensemble, which leads to a higher robustness against black-box attacks and non-adversarial noise. As we will show, `LOTOS` is highly efficient for convolutional layers and is very fast compared to prior methods. We also show that it is an effective method for training robust ensembles of heterogeneous architectures, where other state-of-the-art (SOTA) methods are not applicable. Finally, we investigate how `LOTOS` is able to improve the results of the prior SOTA methods when they are combined. In short, the main contributions of this work are:

**Lipschitz continuity is not as effective for ensemble robustness.** To the best of our knowledge, we are the first to study the adverse effect of Lipschitz continuity on the transferability rate of adversarial examples (§ 3). Prior works study individual model and ensemble robustness separately which does not reveal the issues that the former might cause when used in the form of an ensemble. We show the presence of a trade-off between single model robustness and ensemble robustness as the Lipschitz constant of the individual models changes through empirical analysis. This shows the necessity of special treatment when training ensembles to alleviate this trade-off while benefiting the proven effectiveness of Lipschitz continuity in the robustness of individual models.

**A new ensemble training method for robustness.** We introduce `LOTOS`, a novel ensemble training method to address the aforementioned trade-off. Prior work diversifies the models in the ensemble using either the final outputs (Kariyappa & Qureshi, 2019; Pang et al., 2019), distilled features (Yang et al., 2020), latent representations (Huang et al., 2023), or the vectorized form of all the parameters (Yang et al., 2021). `LOTOS` considers the corresponding affine layers of the models and orthogonalizes them with respect to one another using a novel component in the loss function. Additionally, we theoretically and empirically show that our method is highly efficient for convolutional layers (§ 4). We demonstrate that `LOTOS` improves ensemble robustness against black-box attacks at a nominal increase in robust accuracy (in some cases). Also, it can be combined with prior SOTA in training robust models and ensembles to further boost their robustness (§ 5).

## 2 RELATED WORK

To increase the diversity of the models in the ensemble, Kariyappa & Qureshi (2019) proposed misalignment of the gradient vectors with respect to the inputs. The intuition behind this idea is how several earlier works generate adversarial examples Goodfellow et al. (2014); Kurakin et al. (2018); Papernot et al. (2016b): these methods use the gradient direction with respect to a given input to find the direction that increases the loss function the most. By moving a small amount toward that direction, examples that are similar to the original example but are misclassified by the model are found. In their paper, Kariyappa & Qureshi (2019) hypothesize that when this gradient direction is the same for various models of the ensemble, the common subspace of their adversarial examples will be larger because the loss function behaves similarly around the original data. They use cosine similarity to capture this similarity in the direction; by incorporating that for pairs of models in the loss function, they diversify the models in the ensemble to make it more robust.

Pang et al. (2019) propose a regularizer to increase the diversity in an ensemble by increasing the entropy in non-maximal predictions. Yang et al. (2020) use an adversarial training objective to increase the diversity of the ensemble by making the non-robust features more diverse. Yang et al. (2021) suggest that not only does the misalignment of the gradient vectors of the loss with respect to the inputs matters, but also the Lipschitz constant of the gradients (not parameters) of the loss with respect to the inputs has to decrease and propose a heuristic method to achieve this, which outperforms the prior methods. Zhang et al. (2022) propose a method based on margin-boosting to diversify the models of an ensemble which did not achieve better results than the prior state-of-the-art methods (Yang et al., 2020; 2021). More recent works try to enhance the robustness of the existing methods by incorporating adversarial training against a variety of publicly available models (Sitawarin et al., 2023) or enhance their time complexity by using faster (but weaker) attacks for data augmentation and enforcing diversity in the latent space (Huang et al., 2023).

There is a different line of work on the orthogonality of the affine layers of the deep learning models which focuses on making the transformation of a single layer orthogonal (i.e., its rows and columns become orthonormal vectors). This helps to preserve the gradient norm of the layer and has been shown to improve the stability and robustness of the models (Trockman & Kolter, 2021; Singla & Feizi, 2021; Xu et al., 2022; Singla et al., 2021; Prach & Lampert, 2022; Hu et al., 2023). This notion of orthogonality is different from what we consider in this work; we wish to make the transformation of "corresponding layers" from different models *orthogonal with respect to each other*.

## 3 MOTIVATION

After introducing the notations, we define the notion of transferability rate (§ 3.1) and continue with introducing our conjecture about the effect of Lipschitz continuity on transferability rate (§ 3.2).

**Notation.** Given the domain of inputs $\mathcal{X}$ and $m$ classes $\mathcal{Y} = \{1, 2, \ldots, m\}$, we consider a multi-class classifier $\mathcal{F} : \mathcal{X} \to \mathcal{Y}$ and its corresponding prediction function $f(x)$ which outputs the probabilities corresponding to each class (e.g., the outputs of the softmax layer in a neural network). The loss function for model $\mathcal{F}$ is denoted $\ell_{\mathcal{F}} : \mathcal{X} \times \mathcal{Y} \to \mathbb{R}_+$; it uses the predicted scores from $f(x)$ to compute the loss given the true label $y$ (e.g., cross-entropy loss). The population loss for model $\mathcal{F}$, which we may also refer to as risk, is defined as $R_{\mathcal{F}}(x, y) = \mathbb{E}_x[\ell_{\mathcal{F}}(x, y)]$. When the models are deep neural networks, we refer to a specific layer using superscripts (e.g., $f^{(i)}$ for the $i$-th layer of deep network). A funtion $f(x)$ is $L$-Lipschitz if $\|f(x) - f(x')\|_2 \le L\|x - x'\|_2, \forall x, x' \in \mathcal{X}$. For a matrix $A$, the spectral norm is defined as $\|A\|_2 = \sup_{x \neq 0} \frac{\|Ax\|_2}{\|x\|_2}$. Any affine layer $i$ (e.g., dense layer, convolutional layer) with transformation matrix $A$ is $\|A\|_2$-Lipschitz. When we say layer $i$ (with transformation matrix $A$) has been clipped to a value $C$, this means we ensure $\|A\|_2 \simeq C$.

### 3.1 DEFINITIONS

We begin with the definition of an adversarial attack and then formally define the *transferability rate ($T_{rate}$)* of adversarial examples.

**Definition 3.1** (Attack Algorithm). For a given input/output pair $(x, y) \in \mathcal{X} \times \mathcal{Y}$, a model $\mathcal{F}$, and a positive value $\epsilon$, a targetted attack algorithm $\mathcal{A}_{\mathcal{F}}^{(t)}(x) = x + \delta_x$ minimizes $\ell_{\mathcal{F}}(x + \delta_x, y_t)$ such that $\|\delta_x\|_2 \le \epsilon$. An untargeted attack $\mathcal{A}(x)$ maximizes $\ell_{\mathcal{F}}(x + \delta_x, y)$.

**Definition 3.2** (Transferability Rate). For an untargeted adversarial algorithm $\mathcal{A}_{\mathcal{F}}$ and input space $\mathcal{X}$, we define the transferability rate ($T_{rate}$) of $\mathcal{A}_{\mathcal{F}}(x)$ from $\mathcal{F}$ to another classifier $\mathcal{G}$, as the following conditional probability:

$$T_{rate}(\mathcal{A}_{\mathcal{F}}, \mathcal{F}, \mathcal{G}) = \mathbb{P}_{(x,y) \in \mathcal{X} \times \mathcal{Y}} \left[ \mathcal{G}(\mathcal{A}_{\mathcal{F}}(x)) \neq y \mid \mathcal{F}(x) = \mathcal{G}(x) = y \wedge \mathcal{F}(\mathcal{A}_{\mathcal{F}}(x)) \neq y \right]. \quad (1)$$

For the transferability of a targeted attack algorithm $\mathcal{A}_{\mathcal{F}}^{(t)}$, and target class $y_t$ this definition is:

$$T_{rate}(\mathcal{A}_{\mathcal{F}}^{(t)}, \mathcal{F}, \mathcal{G}) = \mathbb{P}_{(x,y) \in \mathcal{X} \times \mathcal{Y}} \left[ \mathcal{G}(\mathcal{A}_{\mathcal{F}}^{(t)}(x)) = y_t \mid \mathcal{F}(x) = \mathcal{G}(x) = y \wedge \mathcal{F}(\mathcal{A}_{\mathcal{F}}^{(t)}(x)) = y_t \right].$$
$$(2)$$

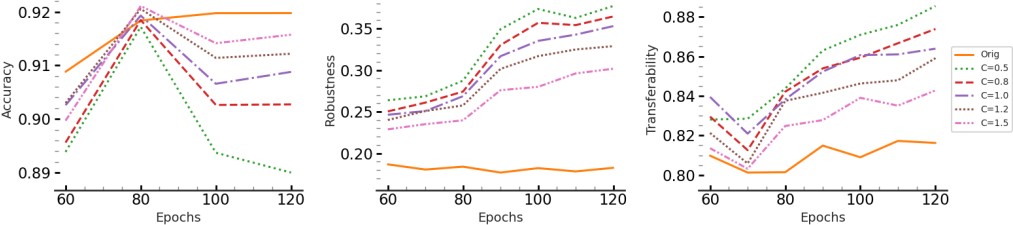

Figure 1: **Accuracy vs. Robust Accuracy vs. Transferability:** Changes in the average accuracy and robust accuracy of *individual* ResNet-18 models, along with the *average transferability rate between any pair of the models in each ensemble* as the layer-wise clipping value (spectral norm) changes. As the plots show, although the robustness of *individual* models increases with decreasing the clipping value, the transferability rate among the models increases, which might forfeit the benefits of the clipping in the robustness of the whole ensemble.

Note that Yang et al. (2021) have similar definitions of transferability except they use joint probabilities instead of conditional probabilities. Using the conditional probability, we can better isolate the "transferability" property we are interested in. By using the joint probability, the $T_{rate}$ will depend on the accuracy of the two models and also the performance of the attack algorithm on the source model $\mathcal{F}$. This will not allow us to have an accurate comparison of the $T_{rate}$ between different settings.

Prior work has shown that there is a trade-off between the accuracy of two models and the $T_{rate}$ of adversarial examples between them (Yang et al., 2021); as the two models become more accurate, their decision boundaries become more similar; this increases the probability that an adversarial example generated for one of the models transfers to the other model because of the similarity of their margins around the source sample. In the following section, by making a parallel to this trade-off, we introduce our conjecture on the trade-off between the Lipschitzness of the models and their $T_{rate}$.

## 3.2 Our Conjecture: Lipschitz Continuity Influences Transferability

Prior works (Szegedy et al., 2013; Farnia et al., 2018; Boroojeny et al., 2024) highlight the importance of Lipschitzness on the robustness of a model against adversarial examples. To enforce the Lipschitzness of the deep neural networks, these works enforce the Lipschitzness on each individual layer: since the Lipschitz constant of the composition of all the components of the model (layers, activation functions, etc.) is upper bounded by the multiplication of their Lipschitz constants, this leads to a bound on the Lipschitzness of the whole model. In this work, we follow the same procedure to control the upper bound on the Lipschitz constant of a model. For this, we use FastClip (Boroojeny et al., 2024) which is the current SOTA method for controlling the spectral norm of dense layers and convolutional layers. We use the chosen spectral norm for each individual layer to represent these models in the figures and tables. For example, $C = 1$ shows that the spectral norm of all the dense layers and convolutional layers have been clipped to 1, which makes each of them 1-Lipschitz. For many architectures such as ResNet-18 (He et al., 2016), and DLA (Yu et al., 2018) this effectively controls the Lipschitz constant of the model because the other constituent components (e.g., ReLU activation, max-pooling, softmax) are 1-Lipschitz (Goodfellow et al., 2014). For details on controlling the effect of batch norm layers, see Appendix B.2.

The prior works have shown that by decreasing the value of the Lipschitz constant $L$ for a model, adversarial attacks achieve a lower success rate. This confirms that decreasing the Lipschitz constant for a model makes it more robust to input perturbations. However, the situation is complicated in an ensemble. While a lower Lipschitz constant promotes robustness for each of the models (of an ensemble), we conjecture that it will increase the $T_{rate}$ by making the classification boundaries of the models more similar. We formalize this intuition below:

**Proposition 3.3.** *Assume $\mathcal{X} = [0, 1]^d$ and $\|\delta_x\| \leq \epsilon$. For two models $\mathcal{F}$ and $\mathcal{G}$, if the loss function on both for any $y \in \mathcal{Y}$ is L-Lipschitz with respect to the inputs, we have the following inequality:*

$$|R_{\mathcal{F}}(\mathcal{A}_{\mathcal{F}}(x), y) - R_{\mathcal{G}}(\mathcal{A}_{\mathcal{F}}(x), y)| \leq 2L\epsilon + |R_{\mathcal{F}}(x, y) - R_{\mathcal{G}}(x, y)|.$$

The proof for the proposition can be found in Appendix A.1. In Proposition 3.3, we study the $T_{rate}$ of an adversarial example generated by $A_{\mathcal{F}}$ to model $\mathcal{G}$ by using the difference in the population loss of

the two models on these adversarial examples as a proxy; the lower this difference is, the more likely that the two models will perform similarly on these adversarial examples. To verify this conjecture empirically, in Figure 1 we show how $T_{rate}$ changes among three ResNet-18 (He et al., 2016) models without batch norm layers as the layer-wise Lipschitz constant (which governs the upper-bound on the Lipschitz constant of the model as mentioned earlier) changes. Figure 5 in Appendix B.3 shows the same behavior for the ResNet-18 models with batch norm layers. For more details see § 5.1. For further discussions on the evidence of this phenomenon in prior theoretical works, see Appendix C.1.

> **Main Takeaway:** According to Proposition 3.3, decreasing $L$ (Lipschitz constant), might be an indicator of higher similarity in the risk of the two models on the adversarial examples, which might imply a higher $T_{rate}$. Therefore, although a lower Lipschitz constant could contribute to the robustness of a single model, it might increase the $T_{rate}$ among the models of an ensemble which might hinder the expected benefits of the Lipschitzness.

## 4  LAYER-WISE ORTHOGONALIZATION FOR TRAINING ROBUST ENSEMBLES

When clipping the spectral norm of the layers, we are reducing the capacity of the parameters that can be used during the optimization (Neyshabur et al., 2017; Bartlett et al., 2017); it is more likely that the parameters of the two models become "more similar" when optimized over these constrained spaces. Therefore, to control the Lipschitz constant of each model to make them more "individually" robust, *and* avoid sacrificing the "ensemble robustness", we need to utilize other modifications to enforce the diversity of their decision boundaries on the adversarially perturbed samples.

> **Our Intuition:** The method that we introduce to enforce the diversity is based on *promoting the orthogonality of the sub-spaces of the corresponding layers of the models that correspond to their top singular vectors*. Since the top singular vectors govern the major part of the transformation by each layer, this orthogonality promotes the difference in the outputs of the corresponding layers from different models.

Affine layers transform the input space such that the sub-space spanned by the top singular vectors will have the most amount of change in the output space for a perturbed input. When the adversary is choosing a perturbation to add to the input, a natural choice would be to choose the direction along the top singular vectors: with the same amount of perturbation, the adversary will get the highest amount of change in the output space for a perturbed input along this direction. Based on this analogy, we consider any two corresponding affine layers $f^{(j)}, g^{(j)}$ from a pair of the models $\mathcal{F}$ and $\mathcal{G}$ in the ensemble, whose linear transformations are represented by matrices $A$ and $B$, with the singular value decompositions $A = \sum_{i=1}^{d} \sigma_i u_i v_i^T$ and $B = \sum_{i=1}^{d} \sigma_i' u_i' v_i'^T$, respectively. We define a notion of similarity based on the top-$k$ sub-spaces:

$$S_k^{(j)}(f^{(j)}, g^{(j)}, \texttt{mal}) := \sum_{i=1}^{k} w_i(\text{ReLU}(\|f^{(j)}(v_i')\|_2 - \texttt{mal}) + \text{ReLU}(\|g^{(j)}(v_i)\|_2 - \texttt{mal})), \quad (3)$$

where (a) $w_i$'s are arbitrary weights which are non-increasing with $i$ to emphasize "more importance" for the singular vectors corresponding to top singular values, and (b) $\texttt{mal}$ refers to the *maximum allowed length* of the output of each layer when it is given the singular vectors of the other layer as the input (see Appendix B.4 for its effect in practice). Observe that when $\texttt{mal}$ is set to 0, the value of $S_k(f, g)$ is 0 if the transformations of $f$ and $g$ are orthogonal in their top-$k$ sub-space (i.e., $\|f^{(j)}(v_i')\|_2 = \|Av_i'\|_2 = \|\sum_{l=1}^{d} \sigma_l u_l v_l^T v_i'\|_2 = 0$).

Utilizing this insight, we introduce our technique, **Layer-wise Orthogonalization for Training Robust Ensembles (LOTOS)**. LOTOS promotes the orthogonality among these sub-spaces which leads to different behaviors when perturbing the clean samples along a specific direction. We add this similarity for each pair of corresponding affine layers (dense and convolutional layers) in each pair of models within the ensemble and add them to the cross-entropy loss. More specifically, given an ensemble of $N$ models $\mathcal{F}_i, \ i = \{1, \ldots, N\}$ with $M$ layers that would be incorporated in the

orthogonalization process, the new loss becomes:

$$\mathcal{L}_{\text{train}} = \frac{1}{N} \sum_{i=1}^{N} \mathcal{L}_{\text{CE}}(\mathcal{F}_i(x), y) + \frac{\lambda}{M\,N\,(N-1)} \sum_{z=1}^{N-1} \sum_{j=z+1}^{N} \sum_{l=1}^{M} S_k^{(l)}(f_z^{(l)}, f_j^{(l)}, \texttt{mal}) \quad (4)$$

where $\mathcal{L}_{\text{CE}}(\mathcal{F}_i(x), y)$ is the cross-entropy loss of $\mathcal{F}_i(x)$ given its output on $x$ and the ground-truth label $y$. $\lambda$ controls the effect of the orthogonalization loss and could be adjusted.

## 4.1 EFFICIENCY OF LOTOS

**Time Complexity:** Note that the number of summands in the orthogonalization loss is $O(N^2 M)$. The computation of $S_k^{(t)}$ uses the computed singular vectors of each layer by FastClip (Boroojeny et al., 2024), which is fast and accurate in practice, and feeds them to the corresponding layer of the other models (see Equation (3)): therefore, it is as if each model has an extra batch of size $N-1$ to process at each iteration, which is relatively small when $N$ is small. In Appendix B.5, we compare the empirical running time per epoch for ensembles of three models. Although this increase in running time is small, our experiments (§ 5) show that performing the orthogonalization for only the first layer would be effective for training robust ensembles (i.e., $M=1$). Therefore, the increase in the training time becomes negligible compared to when the clipping model is used without $\texttt{LOTOS}$.

**Highly Efficient for Convolutional Layers:** For the orthogonalization to be effective in Equation (4), it is necessary to increase the value of $k$ (dimension of orthogonal sub-spaces) because the layers of the DNNs are transformations between high dimensional representations. Therefore, only orthogonalizing the sub-space corresponding to the few top singular values does not guarantee that there is no strong correlation among the remaining top singular vectors (which might correspond to high singular values in each of the models). However, increasing $k$ decreases the computational efficiency, in both compution of the singular vectors (Boroojeny et al., 2024) and computation of the orthogonalization loss in Equation (4).

Fortunately, specific properties of the convolutional layers, which are the most common affine layers in DNNs, allow an effective orthogonalization even with very small values of $k$. In Theorem 4.1, we prove even $k=1$ can be effective in orthogonalization with respect to the remaining singular vectors for convolutional layers.

**Theorem 4.1.** *Given two convolutional layers, $M_1$ and $M_2$ with a single input and output channel and circular padding for which $\boldsymbol{f}$ is the vectorized form of the filter with a length of $T$, and considering $n$ to be the length of the vectorized input, if $\|Av_1'\|_2 \leq \epsilon$, then:*

$$\|Av_p'\|_2 \leq \sqrt{\epsilon^2 + \pi \|\boldsymbol{f}\|_2^2\, T^2 \frac{p}{n}}, \quad (5)$$

*where $A$ is the corresponding linear transformation of $M_1$ and $v_p'$ is singular vector of $M_2$ corresponding to its $p$-th largest singular value.*

The proof can be found in Appendix A.2. As Theorem 4.1 shows, by orthogonalization of the linear transformation of the convolutional layer $M_1$ (i.e., $A$) and only the first singular vector of $M_2$ (i.e., $v_1'$), so that $\|Av_1'\|_2 \leq \epsilon$, the size of the output of $M_1$ when applied to the remaining singular vectors of $M_2$ (i.e., $\|Av_p'\|_2$) will be upper bounded as shown in Equation (5). This upper-bound depends on the ratio of the ranking of the corresponding singular value to the input size (i.e., $\frac{p}{n}$), which gets smaller for the top singular vectors that have a higher contribution to the transformations. It also depends on the size of the kernel ($T$) which is usually small in models used in practice (e.g., $3^2$ in 2D convolutional layers). Finally, it also depends on the $\ell_2$ norm of the filter values, which can be controlled simply by using weight decay when optimizing the parameters during training. We verify this efficiency of $\texttt{LOTOS}$ when applied to convolutional layers in our experiments.

## 5 RESULTS

We wish to answer the following questions: (1) Does decreasing the Lipschitz constant of the models of an ensemble increase the $T_{rate}$ between them?; (2) does $\texttt{LOTOS}$ decrease the $T_{rate}$ among the

models of an ensemble, and does this decrease in the $T_{rate}$ among the models of the ensemble lead to a lower success rate in black-box attacks from other source models?; (3) what are the effects of varying the ensemble size and the number of orthogonalized singular vectors ($k$) on the performance of LOTOS?; (4) is LOTOS still effective when the models of the ensemble are different?; (5) can we combine LOTOS with the prior work on training robust ensembles to provide additional enhancements to robustness?; (6) can LOTOS be combined with common methods used for increasing the robustness of the models, such as adversarial training?; and (7) is LOTOS effective for non-adversarial noise?

As a quick summary, our results show that: (1) decreasing the Lipschitz constant of the models of an ensemble, although make them *individually* more robust, increases the $T_{rate}$ among them (§ 5.1); (2) LOTOS is indeed effective at reducing the $T_{rate}$ between the models of an ensemble which leads to more robust accuracy against black-box attacks (§ 5.2); (3) when using LOTOS, increasing the ensemble size leads to much higher improvement in the robust accuracy (§ 5.3.1), and changing the number singular values has negligible impact on the transferability (§ 5.3.2); (4) LOTOS is effective even when the ensemble is heterogeneous (§ 5.4); (5) LOTOS in conjunction with TRS (Yang et al., 2021) or DVERGE (Yang et al., 2020), two of the SOTA methods in training robust ensembles, yields better performance than either in isolation (§ 5.5); (6) LOTOS can be used together with adversarial training to boost the robustness of the ensemble (Appendix B.8); and (7) by effective diversification of the clipped models, LOTOS enhances robustness against non-adversarial noise.

**Attacks:** We use both black-box attacks and white-box attacks in our experiments. The **white-box** attack is used to evaluate the $T_{rate}$ of adversarial examples between the models in the ensemble; for each ordered pair of the models in the ensemble, the former is used as the source model to generate the adversarial examples and then the $T_{rate}$ of the generated adversarial examples is evaluated on the latter (target model) using Definition 3.2. The average of this value for all the ordered pairs of the models is considered the $T_{rate}$ of the ensemble. A low $T_{rate}$ between the models of the ensemble does not necessarily imply a more robust ensemble. So to evaluate the robustness of ensembles against adversarial attacks, we also use black-box attacks. In the **black-box** attacks, an independently trained source (surrogate) model (of the same type as the models in the ensemble) is used to generate the adversarial examples; we then measure the robust accuracy of the ensembles against these adversarial examples. For further details on the setup of experiments, please refer to Appendix B.1. We also evaluate the robustness against white-box attacks on the whole ensemble (see Appendix B.10.4).

## 5.1 ROBUSTNESS VS. TRANSFERABILITY

In this section, we evaluate our conjecture from § 3.2 which was motivated by Proposition 3.3. For this, we compare the ensemble of three ResNet-18 models trained without any modification (represented as Orig) to ensembles of models in which all the layers of each model is $C$-Lipschitz (by controlling the spectral norm). We vary this Lipschitz constant to see how it affects the robustness and transferability. The chosen Lipschitz constant for each ensemble is used to represent that in results: for example, $C = 1.0$ shows that the affine layers of the models in the ensemble are all clipped to 1.0. The clipping of the layers was achieved using FastClip (Boroojeny et al., 2024).

Figure 1 shows the results when the batch norm layers are removed. The first two subfigures show the changes in the average accuracy and robust accuracy of *individual* ResNet-18 models. The rightmost plot shows the average $T_{rate}$ between any pair of the models in each ensemble as the layer-wise clipping value (spectral norm) changes. As the figure shows, although the robustness of *individual* models increases with decreasing the clipping value, the $T_{rate}$ among the models increases. Figure 5 in Appendix B.3 shows the same behavior for models with batch norm layers (see Appendix B.2 for more details on evaluating these two cases separately).

## 5.2 EFFICACY OF LOTOS

We first evaluate the effectiveness of LOTOS in decreasing the $T_{rate}$ among the clipped models using the white-box attack. For this, we first use ensembles of three ResNet-18 models and follow the setting explained in Appendix B.1. Figure 2 shows the results for 3 different methods of training ensembles (Orig, $C = 1$, and LOTOS). The left-most subfigure shows the average test accuracy of the *individual* models in each ensemble and the middle subfigure shows the average robust accuracy of the *individual* models in the ensemble. The middle plot shows, as expected, that the individual models in both $C = 1$ and LOTOS ensembles are much more robust than the ones in the Orig

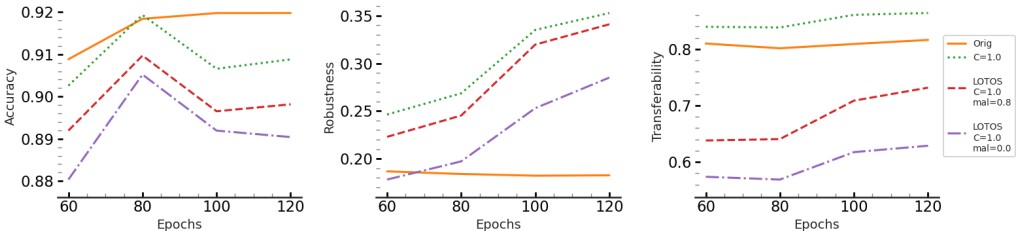

Figure 2: **Reducing transferability while maintaining the benefits of Lipschitzness.** The average test accuracy (left-most plot) and average robust accuracy (middle plot) of the *individual* models in an ensemble of three ResNet-18 models, along with the $T_{rate}$ of adversarial examples between the models of the ensemble (see Appendix B.1). LOTOS keeps the robust accuracy of individual models in the ensemble much higher than those of the Orig ensemble and as mal increases, it becomes more similar to the models in $C = 1$. LOTOS leads to a much lower transferability ($\simeq 20\%$) and the difference increases as mal decreases (right-most plot). These benefits come at a slight cost to the average accuracy of the individual models (left-most plot).

ensembles because of their Lipschitzness property; however, the *individual* models in LOTOS are not as robust as the ones in $C = 1$ because in LOTOS we are enforcing orthogonalization in addition to the Lipschitzness property. Because of this trade-off, as the plot shows, by increasing the value of mal (from 0 to 0.8) the robustness of the individual models becomes more similar to the ones in $C = 1$ ensembles. As the right-most subfigure shows, the $T_{rate}$ between the models in ensembles trained with LOTOS is much lower than $C = 1$ and Orig, and as the mal value decreases (the orthogonalization becomes more strict) the $T_{rate}$ decreases. Figures 9 (Appendix B.7) and 4 show similar effectiveness in reducing the $T_{rate}$ for ensembles that consist of other models (ResNet-18, ResNet-34, and DLA). See § 5.4 for more details on those figures. Also in Appendix B.4 we show how LOTOS effectively performs the orthogonalization of one layer with respect to the corresponding layer in other models of the ensemble while maintaining the target spectral norm.

So far, we observed that LOTOS leads to a noticeable decrease in the transferability at a slight cost in test accuracy and robust accuracy of *individual* models; to make sure that the former overpowers the latter and derives more robust ensembles, we evaluate the robustness of the ensembles against black-box attacks using independently trained surrogate models (see Appendix B.1 for details). We perform this experiment for both ResNet-18 and DLA models and use both CIFAR-10 and CIFAR-100 for the analysis. The results are presented in Table 1. As the table shows, for each choice of the model architecture and dataset, we train ensembles with either of the 3 methods (i.e., Orig, $C = 1$, and LOTOS) and compute their test accuracy and robust accuracy. As the table shows, LOTOS achieves higher robust accuracy in all cases with slight cost to the test accuracy in some cases. As Table 5 in Appendix B.6.3 shows, this difference is even more prominent when the batch norm layers are removed because the clipping algorithms are less accurate (see Appendix B.2).

| | **CIFAR-10** | | | **CIFAR-100** | | |
|---|---|---|---|---|---|---|
| | ORIG | $C = 1.0$ | LOTOS | ORIG | $C = 1.0$ | LOTOS |
| | ENSEMBLES OF RESNET-18 MODELS | | | | | |
| TEST ACC | **95.3 ± 0.06** | 94.7 ± 0.24 | 94.6 ± 0.19 | **77.2 ± 0.17** | 76.6 ± 0.01 | 76.6 ± 0.10 |
| ROBUST ACC | 30.3 ± 1.63 | 35.2 ± 0.72 | **36.3 ± 0.88** | 15.2 ± 0.45 | 18.9 ± 0.40 | **20.2 ± 0.47** |
| | ENSEMBLES OF DLA MODELS | | | | | |
| TEST ACC | **95.4 ± 0.12** | 95.2 ± 0.05 | 95.05 ± 0.09 | 77.1 ± 0.09 | **78.8 ± 0.31** | 78.3 ± 0.38 |
| ROBUST ACC | 26.7 ± 0.58 | 32.8 ± 1.28 | **34.5 ± 0.63** | 16.5 ± 0.78 | 19.4 ± 0.32 | **21.0 ± 0.39** |

Table 1: **Robust accuracy against black-box attacks in ensembles of ResNet-18 models and ensembles of DLA models trained on CIFAR-10 and CIFAR-100** . The surrogate models are a combination of both original models and clipped models trained with multiple random seeds. The target models are ensembles of three models from each architecture choice that are trained using either of the three training methods.

## 5.3 ABLATION STUDIES

In this section, we explore the effect of $k$ when orthogonalizing the top-$k$ sub-spaces of the convolutional layers. We also investigate the effectiveness of LOTOS as the ensemble size increases.

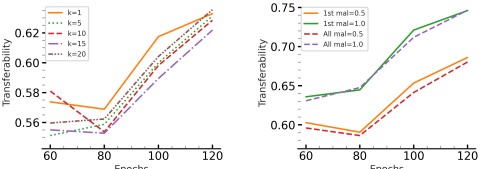

Figure 3: **(Left) Effect of** $k$**.** As the plot shows, the transferability slightly decreases (up to 1%) as $k$ gets larger up to some point (here $k = 15$) and then starts to increase ($k = 20$). **(Right) First layer might be enough!** Comparing the effect of applying LOTOS to only the first layer rather than all the convolutional layers. The similarity of the results motivates the effectiveness of LOTOS for heterogeneous models where it can only be applied to the first layers.

|  | $N = 3$ | $N = 9$ | Improvement |
|---|---|---|---|
| ORIG | $16.4 \pm 1.11$ | $17.0 \pm 1.35$ | 3.7% |
| $C = 1$ | $26.4 \pm 0.57$ | $27.5 \pm 0.95$ | 4.2% |
| LOTOS | $\mathbf{32.2 \pm 0.99}$ | $\mathbf{45.1 \pm 2.28}$ | **40.1%** |

Table 2: **Robust accuracy against black-box attacks in ensembles with different sizes**. The target models are ensembles in three different cases; original models (Orig), clipped ones ($C = 1$), and trained with LOTOS. As the table shows LOTOS has the highest robust accuracy in each ensemble size and it has the highest rate of improvement when the number of models in the ensemble increases from 3 to 9. Note that the robust accuracy of the single Orig model and a single $C = 1$ model are 16.1±0.87 and 26.2±0.52.

In Appendix B.6 we also present additional ablation studies; we evaluate the importance of using the singular vectors in Equation (3) by comparing the transferability when they are replaced with random vectors (Appendix B.6.1). We also evaluate the effect of mal value on the transferability (Appendix B.6.2) and the effect of Lipschitz constant on the strength of the surrogate model in black-box attacks (Appendix B.6.3). For an ablation study on attack methods see Appendix B.10.

### 5.3.1 INCREASING ENSEMBLE SIZE

To verify the effectiveness of LOTOS as the ensemble size increases, we evaluate the improvement in the robust accuracy of the ensembles against black-box attacks (see Appendix B.1 for details) when the number of models in the ensemble increase by factors of 3. Table 2 shows the results of this experiment for ensembles of ResNet-18 models on CIFAR-10 dataset; the improvement in the robust accuracy when the ensemble size increases is relatively small for the Orig and $C = 1$ ensembles. However, when LOTOS is used to train the ensemble, there is a huge improvement in the robustness of the ensemble as the size increases which is due to the increased diversity of the models by orthogonalizing them with respect to one another. The lack of noticeable improvement in the $C = 1$ ensembles highlights the fact that making each model individually robust does not necessarily improve the robustness of the ensemble.

### 5.3.2 DIMENSION OF ORTHOGONAL SUB-SPACES ($k$)

To further evaluate the theoretical observation from Theorem 4.1 for general convolutional layers, we use white-box attacks to measure the $T_{rate}$ among the models in an ensemble of three ResNet-18 models when different values of $k$ are used for orthogonalization of the top-$k$ subspace (see Equation (3)). As Figure 3 (Left) shows, with increasing $k$ there is a slight improvement in the $T_{rate}$ but is still less than one percentage point (compared to $k = 1$) even when $k = 15$. For $k \geq 20$, we noticed a degradation in the training of the models and the $T_{rate}$, which might be due to over-constraining the models. Given the computational efficiency and the negligible difference in the transferability, we found $k = 1$ to be enough and used it in our other experiments.

### 5.4 HETEROGENEOUS ENSEMBLES

Although the original formulation of LOTOS relies on the similarity of the architecture which allows the layer-wise orthogonalization with respect to other models, when considering different architectures, the first affine transformation of each model is applied to the input data, and therefore has still the same vector space for right singular vectors. Therefore, LOTOS can still be utilized to orthogonalize the first layers on different models. Also, as Figure 3 (Right) shows, applying LOTOS to only the first layers would still effectively decrease the transferability among the ResNet-18 models in an ensemble of three models trained on CIFAR-10. In this experiment, we consider ensembles of one ResNet-18, one ResNet-34, and one DLA model on CIFAR-10 and report the average accuracy and robust accuracy of individual models, along with the average transferability among them using white-box attacks. As Figure 4 shows, LOTOS is effective in reducing the transferability of different models

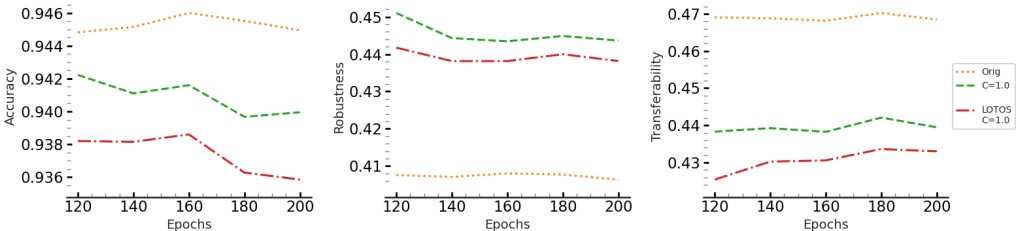

Figure 4: **Investigating the effect of `LOTOS` on the average accuracy and robust accuracy of each of the models of heterogeneous ensembles of DLA, ResNet-18, and ResNet-34 models**, along with presenting the average transferability among any pair of the models in the ensemble as the training proceeds. As the plots show, `LOTOS` leads to a lower transferability among the models while maintaining the benefits of controlling the Lipschitz constant on the robustness of individual models.

by orthogonalizing only their first convolutional layers, as that would lead to different behaviors on the perturbations of the input. Figure 9 in Appendix B.7 shows results for a similar experiment for these models when their batch norm layers are removed. As expected (see Appendix B.2 for details), the improvement in the $T_{rate}$ is much higher without batch normalization layers.

## 5.5 Improving Prior Methods

In this section, we are interested in observing the effectiveness of `LOTOS` when combined with prior SOTA methods for ensemble robustness. Table 3 shows the results for the ensembles trained with TRS (Yang et al., 2021) only (TRS), trained with TRS with clipped models (TRS + $C = 1$), and trained with both TRS and `LOTOS` (TRS + `LOTOS`). As the results show, for both datasets and both model architectures, the robust accuracy of the combined methods is the highest value. This comes at a slight decrease in accuracy (more noticeable for CIFAR-100). Gao et al. (2022) showed that DVERGE (Yang et al., 2020) is more effective than TRS against more elaborate attacks that are specifically designed for ensembles. Hence, we also evaluate the combination of LOTOS with TRS and DVERGE against these attacks to further verify the effectiveness of `LOTOS` (see Appendix B.9).

|  | CIFAR-10 | | | CIFAR-100 | | |
|---|---|---|---|---|---|---|
|  | TRS | TRS + $C = 1$ | TRS + `LOTOS` | TRS | TRS + $C = 1$ | TRS + `LOTOS` |
|  | Ensembles of ResNet-18 models | | | | | |
| Test Acc | $\mathbf{94.4 \pm 0.05}$ | $94.1 \pm 0.17$ | $92.7 \pm 0.09$ | $\mathbf{73.28 \pm 0.46}$ | $72.94 \pm 0.29$ | $67.23 \pm 1.22$ |
| Robust Acc | $30.8 \pm 0.65$ | $35.9 \pm 1.35$ | $\mathbf{41.5 \pm 1.04}$ | $12.3 \pm 0.53$ | $16.3 \pm 0.57$ | $\mathbf{20.7 \pm 0.99}$ |
|  | Ensembles of DLA models | | | | | |
| Test Acc | $\mathbf{94.72 \pm 0.06}$ | $92.79 \pm 0.13$ | $93.18 \pm 0.14$ | $\mathbf{72.6 \pm 0.54}$ | $63.3 \pm 1.20$ | $66.8 \pm 1.26$ |
| Robust Acc | $31.2 \pm 0.80$ | $32.9 \pm 0.77$ | $\mathbf{35.3 \pm 0.39}$ | $23.2 \pm 0.41$ | $23.7 \pm 2.36$ | $\mathbf{24.3 \pm 1.67}$ |

Table 3: **Robust accuracy against black-box attacks in ensembles of ResNet-18 and ensembles of DLA models trained with TRS.** We use ensembles of three models for three different cases; trained with TRS only, trained with TRS while clipping the models, and trained with both TRS and `LOTOS`. As the results show, using TRS and `LOTOS` achieves a robust accuracy that is higher than when either of these methods is used.

## 6 Conclusions

We showed there is a trade-off between the robustness of individual models in the ensemble and the transferability rate of adversarial examples among them as the Lipschitz constant of the models changes. This trade-off prevents the expected boost in the robustness of the ensembles of models when they are Lipschitz continuous. Motivated by this observation, we proposed `LOTOS` that decreases the transferability rate by orthogonalizing the top sub-space of the corresponding layers of different models with respect to one another. We performed a thorough ablation study on the components of our method and showed the effectiveness of `LOTOS` in boosting the robustness of ensembles. We discuss some of the limitations of our proposed method in Appendix C.2.

## 7 ACKNOWLEDGMENT

This work used Delta computing resources at National Center for Supercomputing Applications through allocation CIS240316 from the Advanced Cyberinfrastructure Coordination Ecosystem: Services & Support (ACCESS) program Boerner et al. (2023), which is supported by U.S. National Science Foundation grants #2138259, #2138286, #2138307, #2137603, and #2138296. This paper was generously supported by NSF award IIS-2312561.

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

APPENDIX

A    PROOFS

A.1    PROOF FOR PROPOSITION 3.3

*Proof.*

$$
\begin{aligned}
R_{\mathcal{F}}(\mathcal{A}(x), y) - R_{\mathcal{G}}(\mathcal{A}(x), y) &= \mathbb{E}_x \ell_{\mathcal{F}}(\mathcal{A}(x), y) - \mathbb{E}_x \ell_{\mathcal{G}}(\mathcal{A}(x), y) = \mathbb{E}_x [\ell_{\mathcal{F}}(\mathcal{A}(x), y) - \ell_{\mathcal{G}}(\mathcal{A}(x), y)] \\
&\leq \mathbb{E}_x [\sup_{\|\delta_x\|_2 < \epsilon} (\ell_{\mathcal{F}}(x + \delta_x, y) - \ell_{\mathcal{G}}(x + \delta_x, y))] \\
&\leq \mathbb{E}_x [\sup_{\|\delta_x\|_2 < \epsilon} (\ell_{\mathcal{F}}(x, y) - \ell_{\mathcal{G}}(x, y)) + 2L\|\delta_x\|_2] \\
&\leq \mathbb{E}_x [(\ell_{\mathcal{F}}(x, y) - \ell_{\mathcal{G}}(x, y)) + 2L\epsilon] \\
&= \int_{\mathcal{X}} \ell_{\mathcal{F}}(x, y) - \ell_{\mathcal{G}}(x, y) + 2L\epsilon \, d(x, y) \\
&= 2L\epsilon + R_{\mathcal{F}}(x, y) - R_{\mathcal{G}}(x, y).
\end{aligned}
$$

We can get a similar inequality starting from $R_{\mathcal{G}}(\mathcal{A}(x), y) - R_{\mathcal{F}}(\mathcal{A}(x), y)$ to derive the desired inequality.

□

A.2    PROOF FOR THEOREM 4.1

To prove this theorem, we first prove Lemma A.1 that derives an upper-bound for the difference of the squared values of the largest singular value and any other singular value of a convolutional layer with 1 input and output channel and circular padding. The main intuition behind the proof of Lemma A.1 is that the singular values of convolutional layers depend only on the real parts of the first few powers of roots of unity (Boroojeny et al., 2024). This causes a correlation between the magnitude of the singular values that correspond to the neighboring roots of unity on the real axis.

**Lemma A.1.** *For a convolutional layer with 1 input and output channel and circular padding, if the length of the vectorized input is $n$, then:*

$$
\sigma_1^2 - \sigma_p^2 \leq \pi \|\boldsymbol{f}\|_2^2 \, T^2 \frac{p}{n},
$$

*where $\boldsymbol{f}$ is the vectorized form of the convolutional filter with a length of $T$.*

*Proof.* The rank of the affine transformation of a convolutional layer is *input dimention* $\times$ $\min(c_{in}, c_{out})$ (Sedghi et al., 2018), which in the setting of this lemma is equal to $n$. Boroojeny et al. (2024) showed that if the vectorized form of the kernel is given by $\boldsymbol{f} = [f_0, f_1, \ldots, f_{T-1}]$, then we have the following equation for singular values of this convolutional layer:

$$
s_j^2 = c_0 + 2 \sum_{i=1}^{T-1} c_i \Re(\omega^{j \times i}), \quad j = 0, 1, 2, \ldots, n-1, \tag{6}
$$

where $c_i$'s are defined as:

$$
\begin{aligned}
c_0 &:= f_0^2 + f_1^2 + \cdots + f_{T-1}^2, \\
c_1 &:= f_0 f_1 + f_1 f_2 + \cdots + f_{T-2} f_{T-1}, \\
&\vdots \\
c_{T-1} &:= f_0 f_{T-1}.
\end{aligned}
$$

and $\omega = \exp(2\pi/n)$ is the basic $n$-th root of unity. The order of $s_j$s (in terms of their magnitude) are unknown apriori and depend on the filter values. Let assume that $\sigma_1 = s_j$ corresponds to the largest singular value. Next, we consider an arbitrary $s_t$ and derive the upperbound for $s_j^2 - s_t^2$:

$$s_j^2 - s_t^2 = c_0 + 2\sum_{i=1}^{T-1} c_i \Re(\omega^{j \times i}) - c_0 - 2\sum_{i=1}^{T-1} c_i \Re(\omega^{j \times i}) = 2\sum_{i=1}^{T-1} c_i \left( \Re(\omega^{j \times i}) - \Re(\omega^{j \times i}) \right).$$

Now, we need to bound the terms in the summation. We use the fact that $\omega^z = \exp(2z\pi i/n) = \cos 2z\pi/n + i\sin 2z\pi/n$ and therefore $\Re(\omega^z) = \cos 2z\pi/n$:

$$\Re(\omega^{j \times i}) - \Re(\omega^{j \times i}) \le |\Re(\omega^{j \times i}) - \Re(\omega^{j \times i})| = |\cos\left( \frac{2\pi j \times i}{n} \right) - \cos\left( \frac{2\pi j \times i}{n} \right)|$$

$$= |2\sin\left( \frac{(j+t)\pi i}{n} \right) \sin\left( \frac{(t-j)\pi i}{n} \right)| \le 2|\sin\left( \frac{(t-j)\pi i}{n} \right)|$$

$$\le 2\frac{|t-j|\pi i}{n},$$

where the last two last inequalities are due to $sin(x) \le 1$ and $sin(x) \le x$, respectively. By using this inequality we can write:

$$s_j^2 - s_t^2 = 2\sum_{i=1}^{T-1} c_i \left( \Re(\omega^{j \times i}) - \Re(\omega^{j \times i}) \right) \le 2\sum_{i=1}^{T-1} 2 c_i \frac{|t-j|\pi i}{n}$$

$$= 4\frac{|t-j|\pi}{n} \sum_{i=1}^{T-1} c_i \times i \le 4\frac{|t-j|\pi}{n} \sum_{i=1}^{T-1} i \times \max_i c_i$$

$$= 2(T-1)T\frac{|t-j|\pi}{n} \times \max_i c_i.$$

It is easy to show that $\max_i c_i = c_0$; for example, for $c_1$ we can write $2(c_0 - c_1) = \sum_{i=0}^{T-2}(f_i - f_{i+1})^2 \ge 0 \implies c_0 \ge c_1$. For $c_i$, $i > 1$ a similar justification can be made by only considering the terms that appear in the summands of $c_i$ and using the fact that for the other terms their squared value, which is non-negative, appears in $c_0$. Therefore:

$$s_j^2 - s_t^2 \le 2\frac{\pi c_0 T^2}{n}|t-j|.$$

Now considering the set of indices $\mathcal{I} := \{j \pm 1, \dots, j \pm \lfloor p/2 \rfloor\}$, we know that $s_t$, $t \in \mathcal{I}$ falls within $\pi c_0 T^2 p/n$ radius of $s_j$ (which we assumed to represent $\sigma_1$). Therefore, there are at least $|\mathcal{I}| + 1 = p$ singular values within this radius (including $\sigma_1$). Hence, $\sigma_p$ (the $p$-th largest singular value) should be within this radius as well, and this completes the proof.

$\square$

Note that the bound holds for any arbitrary pair of singular values and can be stated in a more general form, as mentioned in Corollary A.2.

**Corollary A.2.** *For the setting of Lemma A.1, with sorted singular values $\sigma_1, \dots, \sigma_n$, the following inequality holds:*

$$\sigma_j^2 - \sigma_{j+p}^2 \le \pi \|f\|_2^2 T^2 \frac{p+1}{n}, \quad j, j+p \in [1, \dots, n].$$

Now, using Corollary A.2, we can easily prove Theorem 4.1 and show that for two convolutional layers, orthogonalization of only the singular vector corresponding to their largest singular values has a similar orthogonalization effect for the remaining singular vectors and this effect increases for the top singular vectors.

*Proof.* **(Theorem 4.1).** Since $M_1$ and $M_2$ are convolutional layers with circular padding and one input channel and output channel, their linear transformation can be represented by circulant matrices of rank $n$ (dimension of the input), $A_1$ and $A_2$ (Goodfellow et al., 2014). Singular vector matrix of any circulant matrix is equal to the Fourier matrix of size $n$, and the singular vector corresponding to singular value $s_j$ from unordered list of singular values in Lemma A.1 can be written as $\frac{1}{\sqrt{n}}[\omega^{j \times 0}, \omega^{j \times 1}, \ldots, \omega^{j \times (n-1)}]^T$ (Gray et al. (2006) Theorem 3.1). Therefore, the singular vectors of $A_2$ (i.e., $v_i'$s) are simply a different ordering of the singular vectors of $A_1$ (i.e., $v_i$s). Hence, each $\|A_1 v_i'\|_2$ is equal to some singular value $s_j$ of $A_1$. Assuming that $\|A_1 v_1'\|_2 = s_j$ and $\|A_1 v_i'\|_2 = s_{j+i-1}$, by making $\|A_1 v_1'\|_2 \leq \epsilon$ and , using Corollary A.2, we concolude the inequality of interest. $\square$

## B  EXPERIMENTS

In this section, we first provide more details on the setting of our experiments (Appendix B.1) and explain how we control for the effect of batch norm layers in our experiments (Appendix B.2). Then we provide more results on the verification of the trade-off between robustness of individual models and transferability rate among them as the Lipschitz constant changes (Appendix B.3). We show how LOTOS effectively performs the orthogonalization among the layers of the models in an ensemble in Appendix B.4 and present the results on three other ablations studies on our proposed method in Appendix B.6. In Appendix B.7 we provide more results on heterogeneous ensembles, and in Appendix B.8, we will investigate the effectiveness of LOTOS in increasing the robustness of the ensembles when combined with adversarial training of individual models. In Appendix B.9 we provide more experiments to further showcase how LOTOS can be combined with other ensemble robustness methods to drastically improve their robustness. In Appendix B.10 we provide the results of a set of extensive experiments to further evaluate the robustness of our proposed method against different attacks with various settings. Finally, in Appendix B.11 we investigate the advantages of LOTOS in increasing the robustness of model ensembles against non-adversarial noise.

### B.1  SETUP

In this section, we elaborate on the setup of our experiments.

**Compute Infrastructure:** We used NVIDIA A40 GPUs for our experiments except for the experiments in § 5.5 that involved training with TRS method where we used NVIDIA A100 GPUs. Using 32GB of RAM was enough for performing our experiments.

**Datasets and Models:** In all the experiments for evaluating the efficacy of our model, either in isolation or in combination with prior methods, we use both CIFAR-10 and CIFAR-100 datasets (Krizhevsky et al., 2009). The models we use in these experiments consist of ResNet-18 (He et al., 2016), ResNet-34 (in experiment on heterogeneous ensembles), and DLA (Yu et al., 2018)[1]. For more exploratory ablation studies (e.g., effect of the parameter $k$ and ensemble size) we limit the experiments to ResNet-18 models on CIFAR-10.

**Attack Details:** We use both black-box attacks and white-box attacks in our experiments.

○ The white-box attack is used to evaluate the transferability rate of adversarial examples between the models in the ensemble; for each ordered pair of the models in the ensemble, the first one is used as the source model to generate the adversarial examples and then the transferability rate of the generated adversarial examples is evaluated on the second model (target model) using Definition 3.2. The final transferability rate for the ensemble is the average of the transferability rate for all the ordered pairs of the models. To make the results more accurate, we repeat this for 3 different ensembles trained from scratch with different random seeds, and report the average values. The attack performed on the source models is PGD-50 with $\|\epsilon\|_\infty \leq 0.04$, unless stated otherwise. We also evaluate the robustness of ensembles against white-box attacks where all the constituent models are provided to the adversary and report the results in Appendix B.10.4 and Appendix B.9.

○ For the black-box attacks, an independently trained source (surrogate) model (of the same type as the models in the ensemble) is used to generate the adversarial examples; we then measure the robust

---

[1]According to https://github.com/kuangliu/pytorch-cifar they achieve superior results on CIFAR-10.

accuracy of the ensembles against these adversarial examples i.e., robust accuracy is the accuracy on the adversarial samples for which the model correctly predict the original versions. We use both the original models and their clipped version as the surrogate models to generate adversarial examples. To make the results more accurate, we train 3 source models with different random seeds (for each of the two types of surrogate models) and 3 ensembles with different random seeds and compute the average of the robust accuracies over the 18 different cases of choosing a source model and a target ensemble. For both white-box attacks and black-box attacks we use PGD-50 (Madry et al., 2017) attack with $\|\epsilon\|_\infty \le 0.04$ to generate the adversarial examples unless stated otherwise. For these attacks we compute $T_{rate}$ according to Definition 3.2 and report $1 - T_{rate}$ as the robust accuracy against generated adversarial examples for transferability attack.

## B.2 BATCH NORMALIZATION LAYERS

Controlling the Lipschitz constant of the models is more complicated in the presence of batch norm layers. For analyzing the properties of the Lipschitz models, some prior works ignore the batch norm layers (Miyato et al., 2018) and some remove them from the models (Sedghi et al., 2018). Some other works modify the parameters of the batch norm directly to ensure Lipschitzness (Gouk et al., 2021; Senderovich et al., 2022; Delattre et al., 2023). Boroojeny et al. (2024) show that batch norm layers show a compensating behavior when the spectral norm of it preceding convolutional layer is controlled so ignoring them does not help with bounding the Lipschit constant of the model. They also show that modifying the parameters directly leads to poor training and test accuracy. They instead propose an approximate method for controlling the Lipschitz constant of the composition of the convolutional and batch norm layers which works better in practice but still are not exact and as accurate. Therefore, for analyzing the true effect of the Lipschitz constant on transferability rate and ablation studies we perform the experiments on models without their batch norm layers, in addition to performing the same experiments on the efficacy of our method in increasing the robustness of ensembles of the original models (including their batch norm layers) when using the approximate methods for controlling the Lipschitz constant in the presence of batch norm layers.

## B.3 ROBUSTNESS VS. TRANSFERABILITY (CONT.)

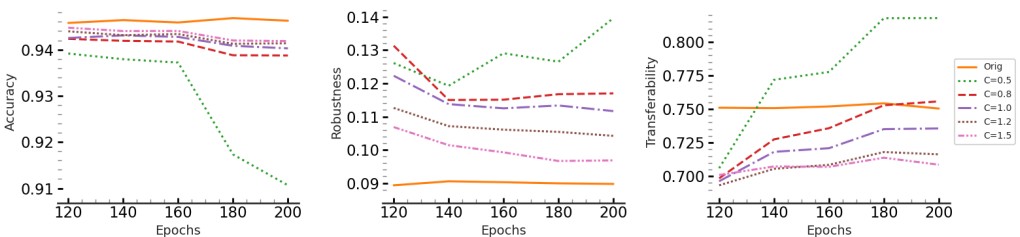

Figure 5: **Accuracy vs. Robust Accuracy vs. Transferability:** Changes in the average accuracy and robust accuracy of *individual* ResNet-18 models (with batch norm layers), along with the average transferability rate between any pair of the models in each ensemble as the layer-wise clipping value changes. As the plots show, although the robustness of *individual* models increases with decreasing the clipping value, the $T_{rate}$ among the models increases, which might forfeit the benefits of the clipping in the robustness of the whole ensemble.

We use the same setting as in § 5.1 for this section, but keep the batch normalization layers in the ResNet-18 models intact. In Figure 1 we observed that LOTOS effectively decreased the transferability rate for models without batch norm layers. In Figure 5 we also see that when the batch norm layers are present, LOTOS is still effective; however, the improvement might not be as much as what was observed without batch norm layers. As mentioned earlier, the reason for this lower effectiveness is that the clipping methods for controlling the Lipschitz constant of the models with batch norm layers are less accurate and not as effective. That leads to a less accurate computation of the singular vectors in Equation (3). Also, batch norm layers are known to adversly affect the robustness of models (Xie & Yuille, 2019; Benz et al., 2021) and prior work has pointed out their compensation behavior when controlling the spectral norm of their preceding convolutional layer (Boroojeny et al., 2024).

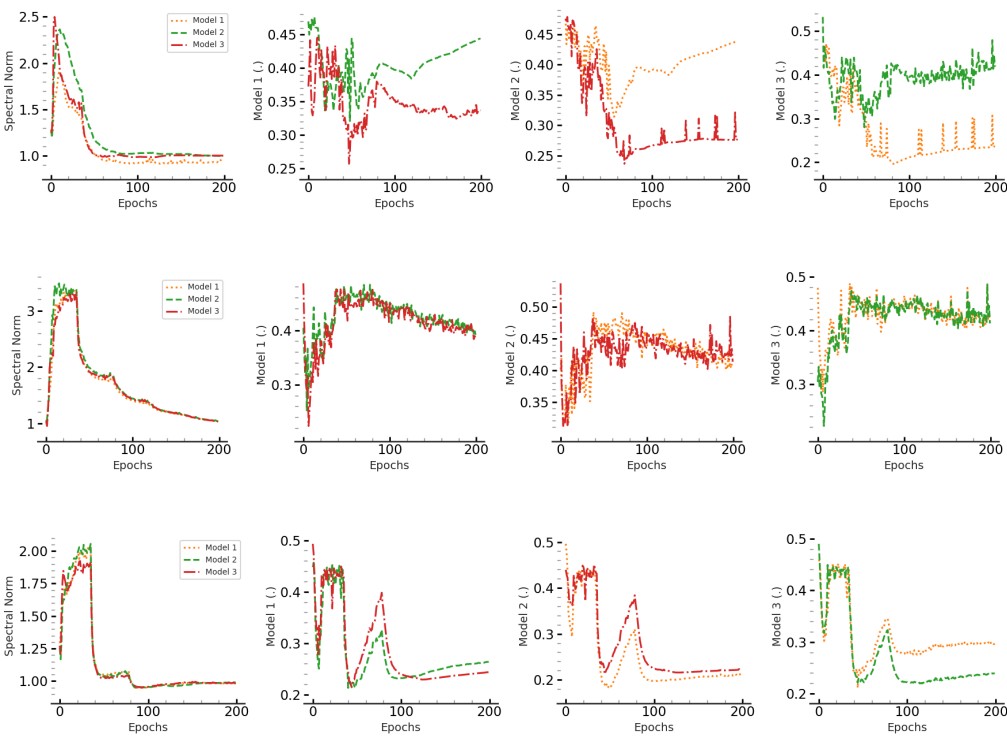

Figure 6: Each row represents the results for a specific layer from the three ResNet-18 models in an ensemble trained with LOTOS with $C = 1$ and mal $= 0.5$. The leftmost subfigure shows the spectral norm of that layer for each model which is enforced to be 1-Lipschitz. The next three subplots show the size of the outputs of that layer from each model when applied to the largest singular vectors of the corresponding layer from the other models; **LOTOS effectively keeps these values below the chosen value for mal**.

## B.4 EFFICACY OF LOTOS (CONT.)

As mentioned previously, the goal of the orthogonalization loss of LOTOS in Equation (4) is to keep the size of the output of a layer from each of the models in the ensemble when applied to the largest singular vector of the corresponding layer from the other models below the chosen mal value. This goal has to be achieved while enforcing the spectral norm of each layer to the target clipping value. In this section, we look at some of the layers of the ResNet-18 models in an ensemble that is trained with LOTOS on CIFAR-10, to see how the aforementioned values change during the training of the model. We randomly choose three of the layers from different parts of the models and evaluate them in the course of training for two different chosen values for mal. Figure 6 shows the results when $C = 1.0$ and mal $= 0.5$. Each row shows the results for one specific layer from the three models. The leftmost subfigure shows the spectral norm of that layer for each of the models. As the figure shows, the clipping method effectively enforces the spectral norm to be almost 1 and therefore makes that layer 1-Lipschitz. The next three subfigures, each show the output of that layer from one of the three models when applied to the largest singular vectors of the corresponding layer from the other two models. As the plots show, these values are effectively controlled to be less than the chosen mal value.

Figure 7 shows the results for the same three layers when the ensemble is trained with mal $= 0.01$. As the plots show the spectral norm is effectively controlled to be almost 1, while the output of the layers on the largest singular vectors of the other layers is made much smaller than in the previous case. However, note that for some of these layers, this size might increase up to $0.1$, which is larger than the mal value and the reason is the specific structure of the convolutional layers which does not allow them to accept arbitrary spectrums as noticed by prior work (Boroojeny et al., 2024).

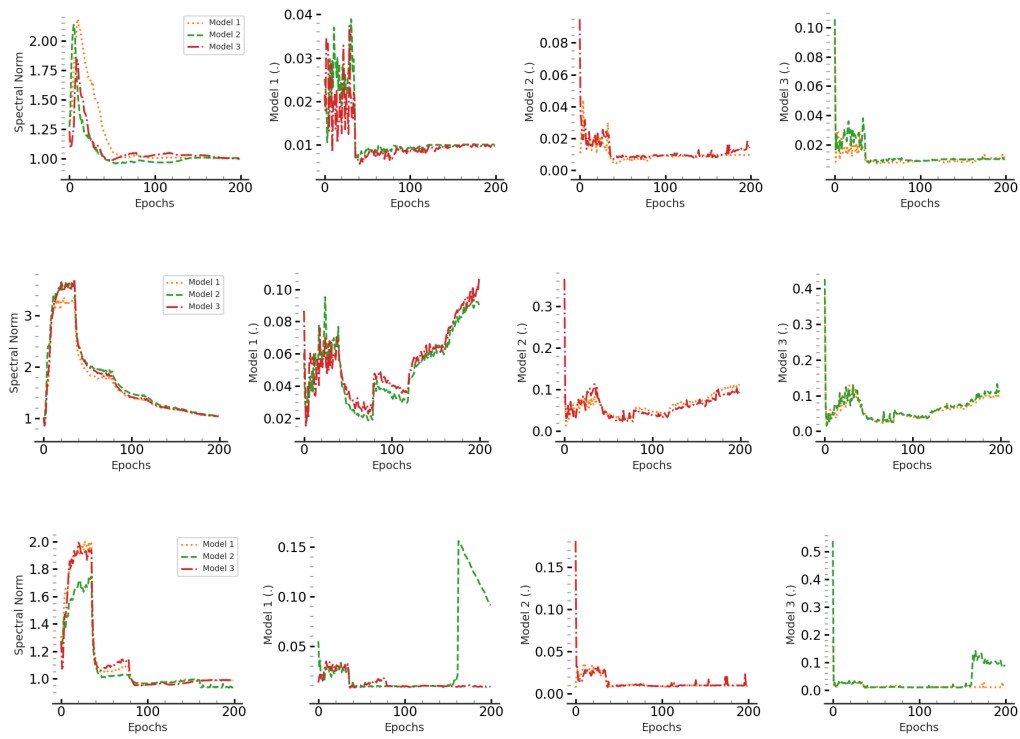

Figure 7: Each row represents the results for a specific layer from the three ResNet-18 models in an ensemble trained with LOTOS with $C = 1$ and $\mathtt{mal} = 0.01$. The leftmost subfigure shows the spectral norm of that layer for each model which is enforced to be 1-Lipschitz. The next three subplots show the size of the outputs of that layer from each model when applied to the largest singular vectors of the corresponding layer from the other models; **LOTOS effectively makes these values much smaller but because the limitations of convolutional layers, they might not become smaller than the `mal` value in some cases**.

## B.5 RUNNING TIME

In § 4.1, we saw that the computation of orthogonalization loss in LOTOS is the same as running each model on $N - 1$ additional batch. Table 4 shows a comparison between the training times of ensembles of three ResNet-18 or DLA models on an NVIDIA A40 GPU. We report the time per epoch for training ensembles without clipping, with clipping, and with LOTOS. For the latter two versions, we also report the increased factor in time compared to the training time of the regular ensembles. These times are shown in three groups: 1. when no additional robustification method is used, 2. when TRS (Yang et al., 2021) is also used for training the ensembles, and 3. when Adversarial Training (Adv) (Madry et al., 2017) is used in training of each individual model in the ensemble. As the table shows the time increase caused by the orthogonalization loss in LOTOS (see Equation (4)) is negligible while being very effective in diversification among the models and decreasing the transferability rate among them.

| | ORIG | $C = 1$ | LOTOS | TRS | TRS + $C = 1$ | TRS + LOTOS | ADV | ADV + $C = 1$ | ADV + LOTOS |
|---|---|---|---|---|---|---|---|---|---|
| RESNET-18 | 33.2 | 74.9 ×2.3 | 79.3 ×2.4 | 158.2 | 224.4 ×1.4 | 227.3 ×1.4 | 312.6 | 479.2 ×1.5 | 485.2 ×1.5 |
| DLA | 63.1 | 155.4 ×2.5 | 165.4 ×2.6 | 326.2 | 466.1 ×1.4 | 477.4 ×1.5 | 758.6.2 | 942.5 ×1.2 | 949.2 ×1.2 |

Table 4: Time (in seconds) per epoch for training ensembles of three ResNet-18 models using either of the methods investigated in this paper. These values are computed on an NVIDIA A40 GPU. As the table shows the orthogonalization loss used in LOTOS makes a negligible change in the computation time of the ensembles with clipped models ($C = 1$).

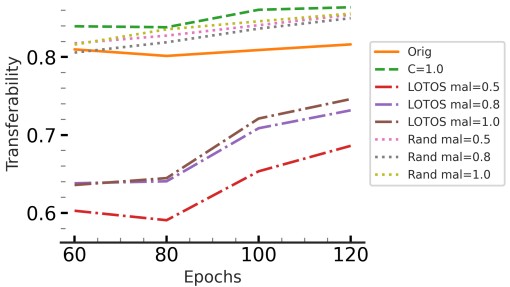

Figure 8: **Do random vectors work?** We use random vectors in Equation (3) instead of the largest singular vectors to verify the effect of orthogonalization on decreasing the transferability. As the plot shows the ensembles that use random vectors rather than the largest singular vectors of the layers of other models perform similarly to the clipped model without `LOTOS` ($C = 1.0$). It still shows slight improvement because using different random vectors still provides some diversity among the models but not as much as using the top singular vectors.

### B.6 ADDITIONAL ABLATION RESULTS

In this section we present the results on three other ablations studies (in addition to the ablation studies presented in § 5.3) on our proposed method; First, we show the importance of using singular vectors of the other layers in Equation (3) by comparing the results when they are replaced by random vectors (Appendix B.6.1). We then investigate the effect of changing `mal` value in Equation (4) in the transferability rate (Appendix B.6.2). Finally, we investigate the effect of changing the clipping value of the source model in black-box attacks (Appendix B.6.3).

### B.6.1 LARGEST SINGULAR VECTOR VS. RANDOM VECTORS

`LOTOS` orthogonalizes the corresponding layers of the models by penalizing the size of the output of each layer when it gets the largest singular vectors of the corresponding layers in the other models as its inputs. To show that the decrease in the transferability rate presented in § 5.2 is indeed the result of orthogonalization of the subspaces spanned by the top singular vectors, we perform an experiment in which we use randomly chosen vectors in Equation (3) instead. As Figure 8 shows, using the transferability rate when random vectors are used is similar to the clipped model without `LOTOS`. The use of random vectors for the models still causes slight decrease in the transferability rate as it introduces new random differences in the training of the models, but not as much as using the top singular vectors which leads to orthogonalization of the models with respect to one another.

### B.6.2 EFFECT OF `MAL`

In Equation (4), by decreasing the maximum allowed length (`mal`) for the size of the output of a layer when top singular vectors of the other layers are given as input, we enforce a higher degree of orthogonalization, and therefore expect to see a more decrease in the transferability rate among the models. As Figure 3 (Right) and Figure 8 show, that is indeed the case. However, based on our experiments, decreasing `mal` to very low values, decreases the robustness and accuracy of individual models. We found the value of $0.8$ to be a good trade-off between the two for increasing the robustness of ensembles and used that for our experiments.

### B.6.3 CHANGING THE LIPSCHITZ CONSTANT OF THE SURROGATE MODEL

In this experiment, we evaluate the effect of Lipschitz constant of theh surrogate model on the effectiveness of its adversarial examples on the target ensembles in a black-box attack. We clip the spectral norm of each layer of each of the source models to a specific value and evaluate the average robust accuracy of each of the three models. We try different layer-wise clipping values ($0.8$, $1.0$, $1.2$, and $1.5$) and for each setting compute the average of the robust accuracy over multiple random seeds. As the table shows the adversarial examples generated on the clipped models are more effective in black-box attacks, but still `LOTOS` has the highes robust accuracy compared to others.

|        | ORIG            | $C = 1.0$        | LOTOS             |
|--------|-----------------|------------------|-------------------|
| ORIG     | $19.3 \pm 1.34$ | $39.0 \pm 0.85$  | $\mathbf{43.8 \pm 1.30}$ |
| $C = 0.8$ | $15.7 \pm 1.03$ | $12.9 \pm 0.64$  | $\mathbf{18.5 \pm 0.79}$ |
| $C = 1.0$ | $13.6 \pm 0.89$ | $13.9 \pm 0.30$  | $\mathbf{20.5 \pm 0.69}$ |
| $C = 1.2$ | $13.2 \pm 0.96$ | $15.9 \pm 0.69$  | $\mathbf{22.9 \pm 0.91}$ |
| $C = 1.5$ | $13.1 \pm 0.65$ | $19.8 \pm 0.29$  | $\mathbf{27.1 \pm 0.66}$ |

Table 5: **Robust accuracy against black-box attacks.** The source models are either original models or clipped models with different layer-wise clipping value for the spectral norm. The target models are ensembles of three ResNet-18 models on CIFAR-10 for three different cases; original models (ORIG), clipped ones ($C = 1.0$), and trained with LOTOS. As the table shows attacks using clipped models are stronger but still LOTOS achieves the highest robust accuracy.

### B.7    HETEROGENEOUS ENSEMBLES (CONT.)

We perform a similar experiment to the one in § 5.4 but with models without batch norm layers. Since the DLA models cannot be trained without batch norm layers, as observed by prior work (Boroojeny et al., 2024), we consider ensembles of one ResNet-18 and one ResNet-34 models on CIFAR-10 and report the average accuracy and robust accuracy of individual models, along with the average transferability among them using white-box attacks. As Figure 9 shows, LOTOS is effective in reducing the transferability of different models by orthogonalizing only their first convolutional layers, as that would lead to different behaviors on the perturbations of the input. As expected, the improvement in the transferability rate is much higher without batch normalization layers (see Appendix B.2).

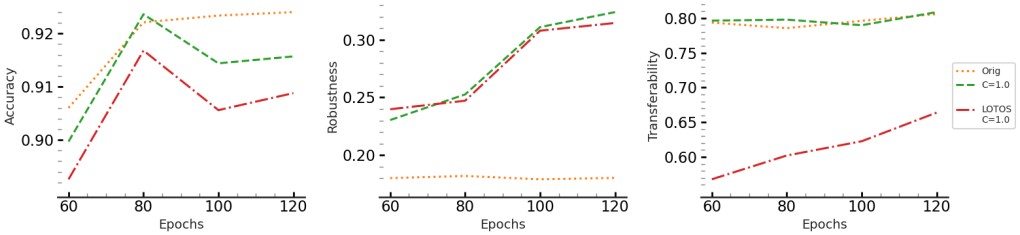

Figure 9: **Investigating the effect of LOTOS on the average accuracy and robust accuracy of each of the models of heterogeneous ensembles of ResNet-18 and ResNet-34 models (without batch norm layers),** along with presenting the average transferability among any pair of the models in the ensemble as the training proceeds. As the plots show, LOTOS leads to a much lower transferability among the models while maintaining the benefits of controlling the Lipschitz constant on the robustness of individual models.

### B.8    LOTOS AND ADVERSARIAL TRAINING

Adversarial training is the most common method for increasing the robustness of individual models in practice. This method finds a new set of adversarial examples for the training set at each iteration of the training algorithm and combines that with the original training data to feed it to the training method for the next iteration. Iterative training of the model on the adversarial examples, which are perturbed versions of the original training samples, makes them less sussptible to the adversarial examples. In this section, we verify that our proposed training paradaigm not only does not interfere with the robustness of the ensembles of models that are adversarialy trained, but also improves the robustness of the ensemble. For this we train ensembles of ResNet-18 and ensembles of DLA models on CIFAR-10 dataset, and incorporate adversarial training for each model within the ensemble. We repeat this procedure for three settings: 1. no further modification (ORIG), 2. clipping each model to 1.0 ($C = 1.0$), and 3. using LOTOS. We evaluate the robust accuracy of these ensembles against blackbox attacks and summarize the results in Table 6. As the table shows, the ensembles trained with LOTOS achieve a higher robust accuracy while achiving similar test accuracy.

|  | ADV TRAIN | ADV TRAIN + $C = 1$ | ADV TRAIN + LOTOS |
|---|---|---|---|
| | ENSEMBLES OF RESNET-18 MODELS | | |
| TEST ACC | $\mathbf{93.1 \pm 0.00}$ | $92.5 \pm 0.16$ | $92.7 \pm 0.09$ |
| ROBUST ACC | $60.1 \pm 0.93$ | $60.9 \pm 1.49$ | $\mathbf{61.7 \pm 1.04}$ |
| | ENSEMBLES OF DLA MODELS | | |
| TEST ACC | $90.8 \pm 0.09$ | $93.7 \pm 0.24$ | $\mathbf{93.9 \pm 0.22}$ |
| ROBUST ACC | $59.6 \pm 0.93$ | $61.2 \pm 1.32$ | $\mathbf{62.7 \pm 1.34}$ |

Table 6: **Robust accuracy against black-box attacks in ensembles of ResNet-18 and ensembles of DLA models trained with adversarial training** We use ensembles of three models for three different cases; trained with adversarial training only, trained with adversarial training while clipping the models, and trained with both adversarial training and LOTOS. As the results show, using both adversarial training and LOTOS achieves a robust accuracy that is higher than when either of these methods are used.

## B.9 IMPROVING PRIOR METHODS (CONT.)

In this section, we evaluate the effectiveness of LOTOS in enhancing the robustness of two SOTA methods, DVERGE (Yang et al., 2020) and TRS (Yang et al., 2021). For this purpose, we evaluate the robustness of different ensembles against MORA (Gao et al., 2022), which is specifically designed for model ensembles and has been shown to be more effective in bypassing the gradient obfuscation introduced by the defense methods (Gao et al., 2022). To evaluate the robustness of ensembles in the worst-case scenario, we choose white-box attacks against these ensembles and use the recommended setting by Gao et al. (2022). For this experiment, we use ensembles of ResNet-18 models trained on CIFAR-10, and for each ensemble robustness method, we use the recommended settings by their authors. As the results in Table 7 show, incorporating LOTOS improves the robust accuracy of TRS and DVERGE ensembles by $4.2$ and $9.5$ percentage points.

We also performed this evaluation for ensembles that are trained only with LOTOS (without TRS and DVERGE), and achieved a robust accuracy of $16.8 \pm 0.51$ against MORA attack. This robust accuracy is higher than TRS but does not reach the robust accuracy of DVERGE alone. However, it is important to note that LOTOS increases the training time of ensembles of ResNet-18 models only by $2.4$ times per epoch (see Appendix B.5) whereas DVERGE increases the training time by a factor of $14.3$ because of all the different techniques they use in their algorithm. If we only add adversarial training to LOTOS, which leads to almost the same (a factor of $14.6$) increase in the training time (see Appendix B.8) for a fair comparison, then the robust accuracy against MORA becomes $49.5\%$ ($\pm 1.45$), which is $29.8$ percentage point higher than DVERGE alone. Note that for all the experiments in this section and section 5.5, we did not perform any hyper-parameter tuning and used all methods with the default parameters.

|  | ORIG | ORIG+ $C = 1$ | ORIG + LOTOS |
|---|---|---|---|
| TRS | $10.3 \pm 0.33$ | $13.2 \pm 1.12$ | $\mathbf{14.5 \pm 0.81}$ |
| DVERGE | $19.7 \pm 2.34$ | $26.8 \pm 0.75$ | $\mathbf{29.2 \pm 0.56}$ |

Table 7: **Boosting robust accuracy of other ensemble robustness methods.** The robust accuracy of ensembles of three ResNet-18 models on CIFAR-10 for two SOTA ensemble robustness methods against MORA white-box attack in three different cases; original models (Orig), clipped ones ($C = 1.0$), and trained in combination with LOTOS. As the table shows LOTOS effectively boosts the robustness of other ensemble robustness methods.

## B.10 OTHER RESULTS

In this section, we perform more experiments to further explore the effectiveness of LOTOS in different settings, including different attack methods, various norm bounds and attack sizes, and white-box attacks. We also evaluate the effectiveness of LOTOS in enhancing the robustness of ensembles of larger models. Finally, we evaluate the effectiveness of our proposed method in improving the ensemble robustness against various non-adversarial noise. For all the experiments

in this section, we use ensembles of ResNet-18 models (without batch norm layers) trained on CIFAR-10.

### B.10.1 CHANGING THE NORM AND RADIUS

In this section, we compare the results when the attack norm is changed in black-box attacks using PGD-50. We first compare the results for $\ell_\infty$ (used in the main experiments), $\ell_2$, and $\ell_1$ norms all with an upper-bound of $0.04$, and report the results in Table 8. The results show the consistent advantage of LOTOS in enhancing the robustness of the ensembles.

|        | $\ell_\infty$      | $\ell_2$           | $\ell_1$           |
|--------|--------------------|--------------------|--------------------|
| ORIG   | $16.4 \pm 1.11$    | $20.4 \pm 1.00$    | $20.2 \pm 0.98$    |
| $C = 1$| $26.4 \pm 0.57$    | $17.9 \pm 1.20$    | $17.9 \pm 1.12$    |
| LOTOS  | $\mathbf{32.2 \pm 0.99}$ | $\mathbf{22.2 \pm 1.79}$ | $\mathbf{22.1 \pm 2.14}$ |

Table 8: **Robust accuracy against black-box attacks with different norms.** The target models are ensembles of three ResNet-18 models on CIFAR-10 for three different cases; original models (ORIG), clipped ones ($C = 1.0$), and trained with LOTOS. As the table shows LOTOS achieves the highest robust accuracy.

We also evaluate the results when the attack size changes for black-box attacks using PGD-50 with $\ell_\infty$ norm bound and report the results in Table 9. As the results show, the benefit of using LOTOS becomes even more clear when the radius of the attack gets larger and the attack becomes stronger.

|        | $\epsilon = 0.02$  | $\epsilon = 0.04$  | $\epsilon = 0.08$  |
|--------|--------------------|--------------------|--------------------|
| ORIG   | $22.9 \pm 1.04$    | $16.4 \pm 1.11$    | $8.2 \pm 0.63$     |
| $C = 1$| $28.2 \pm 1.03$    | $26.4 \pm 0.57$    | $20.7 \pm 0.50$    |
| LOTOS  | $\mathbf{33.1 \pm 1.14}$ | $\mathbf{32.2 \pm 0.99}$ | $\mathbf{28.1 \pm 1.63}$ |

Table 9: **Robust accuracy against black-box attacks with different radii.** The target models are ensembles of three ResNet-18 models on CIFAR-10 for three different cases; original models (ORIG), clipped ones ($C = 1.0$), and trained with LOTOS. As the table shows LOTOS achieves the highest robust accuracy.

### B.10.2 OTHER BLACK-BOX ATTACKS

In this section, we replace PGD-50 with other attack methods to generate the adversarial examples on the surrogate models and evaluate the robust accuracy of target models on the generated examples. For this purpose, we use AutoAttack (Croce & Hein, 2020) and MORA (Gao et al., 2022) attack methods, which are stronger white-box attacks against the surrogate models (see Appendix B.10.4). The results are shown in Table 10. As the results show, although AutoAttack and MORA are stronger attacks in the white-box setting, the adversarial examples they generate for the surrogate models do not generalize to the target ensemble. This is an interesting observation that suggests overfitting the adversarial examples to the surrogate models and needs further exploration in future work. Still, we can see that LOTOS effectively achieves higher robust accuracy against different attacks by diversifying the models within the ensemble and reducing the transferability rate among them.

### B.10.3 LARGER MODELS

To evaluate the effectiveness of LOTOS on more robust models, we evaluated the robustness of ensembles of ResNet-50 models, which are shown to be more robust than the models used in our experiments Peng et al. (2023). We compared the robust accuracy of ensembles of these models against black-box attacks on CIFAR-10. As the results in Table 11 show, LOTOS still effectively achieves a higher robust accuracy by increasing the diversity of the Lipschitz continuous models.

### B.10.4 WHITE-BOX ATTACKS

In this section, we evaluate the robust accuracy of the ensembles against white-box attacks. For these attacks, all the parameters of the models within the ensemble are available to the adversary. Although

|      | PGD-50 | AUTOATTACK | MORA |
|------|--------|------------|------|
| ORIG | $16.4 \pm 1.11$ | $16.3 \pm 0.69$ | $30.0 \pm 0.52$ |
| $C = 1$ | $26.4 \pm 0.57$ | $30.0 \pm 0.39$ | $34.9 \pm 1.07$ |
| LOTOS | $\mathbf{32.2 \pm 0.99}$ | $\mathbf{33.7 \pm 0.76}$ | $\mathbf{40.1 \pm 0.83}$ |

Table 10: **Robust accuracy against black-box attacks with different attack methods.** Different attack algorithms are used to generate adversarial examples on the surrogate ResNet-18 model ensembles. The target models are ensembles of three ResNet-18 models on CIFAR-10 for three different cases; original models (ORIG), clipped ones ($C = 1.0$), and trained with LOTOS. As the table shows LOTOS achieves the highest robust accuracy for all three types of attacks.

|      | RESNET-18 | RESNET-50 |
|------|-----------|-----------|
| ORIG | $16.4 \pm 1.11$ | $27.0 \pm 1.03$ |
| $C = 1$ | $26.4 \pm 0.57$ | $23.7 \pm 1.29$ |
| LOTOS | $\mathbf{32.2 \pm 0.99}$ | $\mathbf{34.8 \pm 1.59}$ |

Table 11: **Robust accuracy against black-box attacks with different model sizes and robustness levels.** The target models are ensembles of three ResNet-18 and ResNet-50 models on CIFAR-10 for three different cases; original models (ORIG), clipped ones ($C = 1.0$), and trained with LOTOS. As the table shows LOTOS effectively boosts the robustness for both model ensembles with different robustness levels.

white-box attacks are considered impractical and unrealistic in most scenarios (Sitawarin et al., 2023), still they can be used as an evaluation for the worst-case scenario. Other than the PGD-50 attacks we used for transferability attacks, we use AutoAttack (Croce & Hein, 2020), which is shown to be a stronger white-box attack, and MORA (Gao et al., 2022), which is specifically designed for model ensembles. As the results in Table 12 shows, LOTOS effectively enhances the robust accuracy of ensembles against white-box attacks.

|      | PGD-50 | AUTOATTACK | MORA |
|------|--------|------------|------|
| ORIG | $21.5 \pm 0.51$ | $18.9 \pm 0.48$ | $18.2 \pm 0.48$ |
| $C = 1$ | $37.4 \pm 0.37$ | $31.5 \pm 0.41$ | $31.0 \pm 0.51$ |
| LOTOS | $\mathbf{39.9 \pm 0.71}$ | $\mathbf{33.3 \pm 0.70}$ | $\mathbf{32.4 \pm 0.43}$ |

Table 12: **Robust accuracy against different white-box attacks.** The target models are ensembles of three ResNet-18 models on CIFAR-10 for three different cases; original models (ORIG), clipped ones ($C = 1.0$), and trained with LOTOS. As the table shows LOTOS achieves the highest robust accuracy against different attack methods.

## B.11    NON-ADVERSARIAL NOISE

In this section, we evaluate the benefits of LOTOS in robustness against various non-adversarial noise types. For this purpose, we use the benchmark introduced by Hendrycks & Dietterich (2019) for CIFAR-10 and report the results in table 13. As the results show, the ensemble of Lipschitz continuous models ($C = 1$), is more robust than the original models by preventing the small noises from drastic changes in the output of the models. However, LOTOS achieves even better robustness by further diversifying these models and preventing different models in the ensemble from failing for the same noise in the inputs.

|       | Gaussian noise | Gaussian blur | Impulse noise | Motion blur | Pixelate | Fog | Frost | Saturate | Shot noise | Zoom blur | Spatter | Speckle noise | AVG |
|-------|----------------|---------------|---------------|-------------|----------|-----|-------|----------|------------|-----------|---------|---------------|-----|
| Orig | 36.1 | 55.7 | 27.0 | 71.2 | 44.3 | **77.5** | 62.5 | **87.4** | 42.9 | 71.8 | 73.5 | 45.8 | 58.0 |
| $C = 1$ | 56.0 | 63.7 | 37.8 | 71.6 | 69.0 | 64.9 | 72.8 | 84.6 | 60.1 | **74.1** | 77.6 | 58.9 | 65.9 |
| LOTOS | **61.6** | **63.8** | **44.5** | **71.8** | **70.8** | 63.3 | 74.8 | 84.4 | **64.5** | 73.8 | **78.2** | **62.7** | **67.9** |

Table 13: **Accuracy against non-adversarial noise.** Accuracy of different ensembles of ResNet-18 models in the presence of different noise types on CIFAR-10. LOTOS effectively achieves higher robustness in most cases and also on average.

## C   FURTHER DISCUSSIONS

In this section, we expand our discussions on the evidence of the trade-off of robustness and transferability due to the Lipschitz continuity in prior theoretical works. We then mention the limitations of our proposed solution (`LOTOS`), to this trade-off.

### C.1   RELEVANT THEORETICAL OBSERVATIONS

In this section, we investigate other theoretical results from prior work that further approve our observations regarding the trade-off of the robustness of single models and transferability rate due to Lipschitz continuity. Theoretical results by Yao et al. (2024), show there is a trade-off between complexity and diversity for transferability rate of adversarial examples. On the other hand, enforcing Lipschitz continuity works as a regularizer to decrease the complexity (Bartlett et al., 2017). So based on the trade-off established by Yao et al. (2024), this decrease in the complexity by Lipschitz continuity leads to reduced diversity among the ensemble models, which in turn reduces the transferability rate. That is why we need an additional modification, such as `LOTOS`, to further boost the diversity despite the reduced complexity caused by Lipschitz continuity. Also, the diversity term defined by Yao et al. (2024) can effectively be increased by `LOTOS` as it directly enforces the differences in the outputs for various perturbations in the inputs.

Wang & Farnia (2023) show that adversarial examples generated by models with controlled operator norm generalize better, which matches our observations (e.g., Figure 1 and Table 5), and motivates the necessity of further training strategies, such as `LOTOS`, to counteract this improved generalization which potentially leads to higher transferability rates. The lower-bounds on the probability of transferability in Wang & Farnia (2023) (similarly in Yang et al. (2021)) show the connection with the smoothness of the model in terms of the Lipschitz continuity of the gradients (not Lipschitz continuity of the outputs which is a milder assumption). Finally, Abe et al. (2023) motivates avoiding the extension of basic intuitions and observations for simple models to neural networks when they are used in the form of an ensemble, as it might not lead to the expected benefits one expects from an ensemble of models. This aligns with our observations (e.g., Table 2) and further motivates a full understanding of the robustness techniques that are designed for individual neural network models before using them in ensembles because they not only might not help in the ensemble settings but may also worsen the results in some cases (Gao et al., 2022).

### C.2   LIMITATIONS OF `LOTOS`

Although we showed the negligible computational overhead of our model, we should mention that it is only limited by the speed of the underlying method used for controlling the spectral norm of affine layers, which has become practical using recent methods (Boroojeny et al., 2024). Another limitation of our method is that it is affected by the degradation of the clipping methods in the models with batch normalization layers. Finally, we should mention that although `LOTOS` was shown to be effective in practice, as we discussed in Section 4.1, it is highly efficient for convolutional layers, but might not be as efficient for other affine layers and we leave the exploration of more efficient variants for other layers to future work.

