# OpenReview forum: "Training Robust Ensembles Requires Rethinking Lipschitz Continuity"
_ICLR.cc/2025/Conference — ICLR 2025 Poster_

### Official Review · Reviewer_HKeX · 2024-10-18

**Soundness:** 3
**Presentation:** 2
**Contribution:** 2
**Rating:** 6
**Confidence:** 3

**Summary:**

The paper studies how Lipschitz continuity affects the transferability of adversarial examples in model ensembles. It finds that while lower Lipschitz constants improve individual model robustness, they increase transferability in ensembles. To address this, the authors propose LOTOS, a training method that promotes subspace orthogonality between models, enhancing ensemble robustness. Experiments show that LOTOS improves the robust accuracy of ResNet-18 ensembles and boosts the performance of existing robust ensemble methods.

**Strengths:**

- It's interesting to observe the trade-off between the robustness of individual models and the transferability of adversarial examples when changing Lipschitz constant. It highlights the importance of addressing the trade-off, an issue that has been overlooked in prior research.
- The proposed LOTOS method is presented as a solution that uniquely focuses on orthogonalizing affine layers, setting it apart from existing methods that target different aspects of model diversity. To my best knowledge, such contribution is also novel.
- The empirical results are shown clearly and shed light on so much understanding of the proposed solution.

**Weaknesses:**

- **The soundness of the claim in this paper**: The authors suggest that "although a lower Lipschitz constant increases the robustness of a single model, it is not as beneficial in training robust ensembles as it increases the transferability rate of adversarial examples across models in the ensemble." While I believe that the first part is sound, I want to challenge the statement "a lower Lipschitz constant is not as beneficial in training robust ensembles as it increases the transferability rate of adversarial examples across models in the ensemble". There are too many aspects that may influence the result, for instance, the generalization of the models (because a lower Lipschitz constant may make the models more difficult to train). In other words, on a broad view, it's not just a trade-off between two aspects mentioned in this paper. I encourage the authors to incorporate deeper understanding of this issue (see below).
- **Missing related work and discussion**: As far as I know, there may be some papers that is relevant to yours. For example, some theoretical works [1,2,3] discuss the smoothness & Lipschitz constant in transferability of adversarial examples (see the bounds therein), which may be somehow relevant to Lipschitz continuity emphasized in this paper. Some of them may not discuss the model ensemble issue. However, these insights on individual models is theoretically connected with the analysis in this paper. Furthermore, the vulnerability-diversity decomposition [4] also sheds light on the relationship among transferability, single model robustness and diversity, which may be in line with your claim "We observe that while decreasing the Lipschitz constant makes each model of the ensemble individually more robust, it makes them less diverse and consequently increases the transferability rate among them which in turn hinders the overall ensemble robustness". In addition, the model ensemble literature [5] also suggests the trade-off between diversity and individual model performance. Therefore, maybe you could give a brief discussion of them to help us better understand your theoretical insights. You are also encouraged to choose to discuss other work that I haven't mentioned but is relevant to your paper.
- **Experiments**: Based on the above, I encourage the authors to incoporate more experiments. Since the model generalization / optimization process may also change the trade-off mentioned in this paper, you may add more experiments regarding more algorithms to show the soundness and consistency of your claim. However, in the current version, the experimental parts only consider their own method as well as TRS [1].

If the author's answers address my concerns, I will consider raising the score.

[1] Yang Z, Li L, Xu X, et al. Trs: Transferability reduced ensemble via promoting gradient diversity and model smoothness[J]. Advances in Neural Information Processing Systems, 2021, 34: 17642-17655.

[2] Zhang Y, Hu S, Zhang L Y, et al. Why does little robustness help? a further step towards understanding adversarial transferability[C]//2024 IEEE Symposium on Security and Privacy (SP). IEEE, 2024: 3365-3384.

[3] Wang Y, Farnia F. On the role of generalization in transferability of adversarial examples[C]//Uncertainty in Artificial Intelligence. PMLR, 2023: 2259-2270.

[4] Yao W, Zhang Z, Tang H, et al. Understanding Model Ensemble in Transferable Adversarial Attack[J]. arXiv preprint arXiv:2410.06851, 2024.

[5] Abe T, Buchanan E K, Pleiss G, et al. Pathologies of predictive diversity in deep ensembles[J]. arXiv preprint arXiv:2302.00704, 2023.

**Questions:**

- What is the definition of $\Re$ in the appendix (page 14)?
- Typos: line 333: "ensebmels"
- Do you think the method in this paper requires significant work on hyperparameter tuning?

---

> ### Author Response · Authors · 2024-11-17
>
> We thank the reviewer for their insightful comments. Below are our responses to their questions and concerns:
>
> ### **1.** The soundness of the claim in this paper and Missing related work and discussion:
>
> We want to emphasize that we do not claim that Lipschitz continuity does not help with making the ensembles more robust (they are indeed more robust than ensembles without any modification). However, the increased transferability rate might hinder their effectiveness, which shows itself in results such as the ones presented in Table 2.
>
> This, in fact, can be explained by theoretical results in the paper [4] you mentioned. That paper has become available **after submitting this work**, but we believe the connections to their result can improve the discussion of our paper. Similar to their theoretical observation, there is a trade-off between complexity and diversity. Enforcing Lipschitz continuity works as a regularizer to decrease the complexity but decreasing the complexity leads to reduced diversity. That is why we need an additional modification, such as LOTOS, to further boost the diversity despite the reduced complexity caused by Lipschitz continuity. There might be other better methods for dealing with this trade-off, but we believe LOTOS will set a reasonable base-line for that. The connections between how LOTOS increases diversity and the diversity term introduced in [4] will also be useful for our discussion in the revised version of our manuscript.
>
> Paper [3] would be a great addition to our discussion to justify the better generalization of adversarial examples generated by models with controlled operator norm, which completely matches our observations (e.g., Figure 1), and motivates the necessity of further training strategies, such as LOTOS, to counteract this improved generalization which potentially leads to higher transferability rates.
>
> The lower-bounds on the probability of transferability in [2] (similarly in [1]) show the connection with the smoothness of the model in terms of the Lipschitz continuity of the gradients (not Lipschitz continuity of the outputs which is a milder assumption), but we believe their results are aligned with our observations and will be good additions to our discussions.
>
> We think [5] is an insightful work that motivates avoiding the extension of basic intuitions and observations for simple models to neural networks when they are used in the form of an ensemble, as it might not lead to the expected benefits one expects from an ensemble of models. This aligns with our observations (e.g., Table 2) and further motivates a full understanding of neural network models when used in ensembles to find better strategies that are worth the additional resource requirements for training these models.
>
> We would like to thank the reviewer for pointing us to the listed papers. We believe all the mentioned papers are useful in our work and improve our motivation and discussion sections.
>
>
> ### **2.** Experiments:
>
> We are running experiments for adding other ensemble training methods (DVERGE). We will post the results here once they are ready. We also added results for other attack methods and $p$-norms as well as other types of noise (blur, pixelated, etc.)  (please see the response to reviewer 1).
>
> ### **3.** What is the definition of $\Re$ in the appendix (page 14)?
>
> We used it to denote the real part of the complex numbers. We will add to our notations. Thanks!
>
>
> ### **4.** Do you think the method in this paper requires significant work on hyperparameter tuning?
>
> For the hyper-parameters of the models and the training of the ensemble (learning rate, optimizer, scheduler, etc.) we used all the default hyper-parameters of these repositories, as they are, in our experiments:
>
> [1] https://github.com/kuangliu/pytorch-cifar
> [2] https://github.com/AI-secure/Transferability-Reduced-Smooth-Ensemble
>
> For the hyper-parameters introduced in LOTOS loss ($mal, k, w_i$), after experimenting with the effect of $k$ and noticing that a value of $1$ is enough, we performed all our experiments with $k=1$, which automatically sets $w_1=1$ (no other $w_i$). For the ablation study on $mal$, we tried different values (Appendix B.6.2), but for all other experiments, we did not do hyper-parameter tuning and fixed it as $0.8$. We think the results can be improved by tuning these hyper-parameters but we did not invest the time as we think we got the results that showcase the effectiveness of our method.

---

> > ### Author Response · Authors · 2024-11-21
> >
> > Regarding additional experiments, we summarize the results that were requested by the other reviewers here, as well:
> >
> > We performed additional experiments using AutoAttack [1] as well as MORA [2] (which is specifically designed for white-box attacks on ensembles) to generate the adversarial examples on the surrogate models and used Definition 3.2 to compute the robust accuracy against transferability-based attacks (similar setting to the PGD-50 with $\ell_\infty \leq 0.04$ reported in Table 2).
> >
> > |               | PGD-50 | AutoAttack | MORA|
> > | ----         | ----------- | ----------- | ------------ |
> > | Orig       | $16.4 \pm 1.11$ | $16.3 \pm 0.69$| $30.0 \pm 0.52$ |
> > | C=1       | $26.4 \pm 0.57$ | $30.0 \pm 0.39$ | $34.9 \pm 1.07$ |
> > | LOTOS | $32.2 \pm 0.99$ | $33.7 \pm 0.76$ | $40.1 \pm 0.83$ |
> >
> >
> > To evaluate our method on other types of noise, we used the benchmark introduced by [3] for CIFAR-10 and report the results in the table below:
> >
> > | | Gaussian noise | Gaussian blur| Impulse noise| Motion blur| pixelate| fog| frost| saturate| Shot noise| Zoom blur| spatter| Speckle noise| **AVG** |
> > | ----         | ----------- | ----------- | ------------ | ---- | ---- | ---- | ---- | ---- | ---- |---- |---- |---- |---- |
> > | Orig       | 36.1 | 55.7| 27.0 | 71.2 | 44.3 |  **77.5** | 62.5 |  **87.4** | 42.9 | 71.8 | 73.5 | 45.8 | 58.0 |
> > | C=1       | 56.0 | 63.7 | 37.8 | 71.6 | 69.0 | 64.9 |72.8 | 84.6 | 60.1 |  **74.1** | 77.6 | 58.9 | 65.9 |
> > | LOTOS |  **61.6** |  **63.8** | **44.5** | **71.8**|  **70.8** | 63.3 |  **74.8** | 84.4 |  **64.5** | 73.8 |  **78.2** | **62.7** | **67.9** |
> >
> > LOTOS ensembles are more robust than the other two because, in addition to benefiting from the Lipschitz continuity introduced in C=1 ensembles, they further diversify the models for how to react to changes in the input.
> >
> > Here, we have performed some experiments with other attack sizes, as well as other norms ($\ell_2$ and $\ell_1$). As we can see with larger attack sizes the effectiveness of our method becomes more clear.
> >
> > Results for PGD-50 with different attack sizes:
> >
> > |               | $\epsilon=0.02$ | $\epsilon=0.04$ | $\epsilon=0.08$ |
> > | ----         | ----------- | ----------- | ------------ |
> > | Orig       | $22.9 \pm 1.04$ | $16.4 \pm 1.11$ | $8.2 \pm 0.63$ |
> > | C=1       | $28.2 \pm 1.03$ | $26.4 \pm 0.57$ | $20.7 \pm 0.50$ |
> > | LOTOS | $33.1 \pm 1.14$ | $32.2 \pm 0.99$ | $28.1 \pm 1.63$ |
> >
> > Results for PGD-50 with $\ell_1 \leq 0.04$ and  $\ell_2 \leq 0.04$:
> >
> > |               | $\ell_\infty$ | $\ell_2$ | $\ell_1$ |
> > | ----         | ----------- | ----------- | ------------ |
> > | Orig       | $16.4 \pm 1.11$ | $20.4 \pm 1.00$ | $20.2 \pm 0.98$ |
> > | C=1       | $26.4 \pm 0.57$ | $17.9 \pm 1.20$ | $17.9 \pm 1.12$ |
> > | LOTOS | $32.2 \pm 0.99$ | $22.2 \pm 1.79$ | $22.1 \pm 2.14$ |
> >
> > We also report the results for ensembles of larger models (ResNet50 ensembles) that are reported to be more robust [4]:
> >
> > |               | ResNet18 | ResNet50 |
> > | ----         | ----------- | ----------- |
> > | Orig       |  $16.4 \pm 1.11$  | $27.0 \pm 1.03$ | l
> > | C=1       |  $26.4 \pm 0.57$  | $23.7 \pm 1.29$ |
> > | LOTOS  |  $32.2 \pm 0.99$  | $34.8 \pm 1.59$ |
> >
> > We will keep you posted as the results of our new experiments become available. Meanwhile, please let us know if you have more questions or concerns. Thanks!
> >
> >
> > [1] Croce et al., Reliable evaluation of adversarial robustness with an ensemble of diverse parameter-free attacks. ICML 2020.
> >
> > [2] Yu et al., MORA: Improving Ensemble Robustness Evaluation with Model-Reweighing Attack. NeurIPS 2022.
> >
> > [3] Hendrycks, D., & Dietterich, T. (2018, September). Benchmarking Neural Network Robustness to Common Corruptions and Perturbations. In International Conference on Learning Representations.
> >
> > [4] Peng, S., Xu, W., Cornelius, C., Li, K., Duggal, R., Chau, D. H., & Martin, J. (2023). Robarch: Designing robust architectures against adversarial attacks. arXiv preprint arXiv:2301.03110.

---

> ### Comment · Reviewer_HKeX · 2024-11-22
> **Response to Authors**
>
> I really appreciate the authors for their hard work, especially the extensive experiments comparing to other methods (as suggested by other reviewers). I'm also happy to see LOTOS perform better than others. My major concerns have been addressed.
>
> While some reviewers have considered the empirical part, I have a question for the theoretical part (the proof of Proposition 3.3, which is a core theoretical part in your motivation). Please let me know how the fourth line is derived in Appendix A.1, i.e.,
> $$\mathbb{E}\_x\left[\sup \_{\|\delta\|_2<r}\left(\ell\_{\mathcal{F}}(x, y)-\ell\_{\mathcal{G}}(x, y)\right)+2 L\|\delta\|_2\right] \leq \mathbb{E}\_x\left[\left(\ell\_{\mathcal{F}}(x, y)-\ell\_{\mathcal{G}}(x, y)\right)+2 L r\right].$$
> I'm not sure why $\sup\_{\|\delta\|_2<r}$ has been cancelled.

---

> > ### Author Response · Authors · 2024-11-22
> >
> > We appreciate the reviewer for checking our new results and we are happy that your major concerns have been addressed!
> >
> > Regarding the derivation of the 4th line in the proof of proposition 3.3, note that by linearity of expectation, we can write:
> >
> > $\mathbb{E}_x [\mathrm{sup}_{\|\delta\|_2 < r} (\ell_\mathcal{F}(x,y) - \ell_\mathcal{G}(x,y)) + 2L\|\delta\|_2] = \mathbb{E}_x [\mathrm{sup}_{\|\delta\|_2 < r} (\ell_\mathcal{F}(x,y) - \ell_\mathcal{G}(x,y))] +  \mathbb{E}_x [2L\|\delta\|_2]$
> >
> > Now note that $\mathbb{E}_x [2L\|\delta_x\|_2] = 2L \mathbb{E}_x [\|\delta_x\|_2] \leq 2Lr$, where the last inequality is due to the assumption of the proposition ($\|\delta\|_2 \leq r$) that considers an upper-bound of $r$ for the perturbation size that the attack algorithm can make, i.e., the attack budge (in the definition of the proposition $\epsilon$ has been used instead of $r$ and we will update the proof to make is consistent. Sorry for the confusion!).
> >
> > Given that you are happy with the experimental results and our theoretical explanation, we hope you will increase your score to support our work.

---

> > > ### Author Response · Authors · 2024-11-22
> > >
> > > sorry, we missed the main part of your question because it was below the scroll bar!
> > >
> > > the reason that $\sup_{\|delta\|_2 < r}$ is canceled is that the term in the parentheses does not depend on $\delta$ anymore. In line 3 we have extracted the delta term from $\ell\mathcal{F}(x+\delta,y) - \ell\mathcal{G}(x+\delta,y)$ and instead added the maximum possible difference that can be upper-bounded by the Lipschitz constant of the loss function ($L$). Note that: $\ell\mathcal{F}(x+\delta,y) \leq \ell\mathcal{F}(x,y) + L \|\delta\|_2$ and $\ell\mathcal{G}(x+\delta,y) \leq \ell\mathcal{G}(x,y) + L \|\delta\|_2$ by the assumption that $\ell\mathcal{F}$ and $\ell\mathcal{G}$ are $L$-lipschitz.
> > >
> > > Thanks for raising score. given the competitive nature of ICLR, a score of 6 suggests that the paper is borderline. if you would like to see the paper published, we kindly request that to reflect in your score, and provide a clearer signal to the AC/SAC.

---

> ### Comment · Reviewer_HKeX · 2024-11-22
> **Thanks**
>
> Thank you for your detailed response! It seems I had some misunderstandings about the derivation. I’ll raise my score to 6, good luck!~

---

> ### Comment · Reviewer_HKeX · 2024-11-22
> **Another concern regarding the novelty of Proposition 3.3 in this paper**
>
> Dear authors,
>
> I deeply appreciate your work, especially the experimental contributions. However, I regret to note that the novelty of the proof in Appendix A.1 appears to be quite limited, leading me to believe that it provides relatively little new insight. **The motivation of this paper is about Lipschitz constant, which may be considered by [1] (see below). In other words, the motivation in this paper has already been hinted at by previous work [1]. That is the reason that why I do not raise the score anymore.**
>
> Specifically, the proof relies on the global Lipschitz constant, which closely resembles the first-order Taylor expansion. However, [1] considers second-order Taylor expansion. Also, since the definition of transferability in this paper is similar to that in [1], a more comprehensive discussion of [1] may be warranted. I have several questions and comments:
> - **Motivation for Lipschitz continuity and its comparison to [1]**: Proposition 3.3 provides a strong motivation to investigate the role of Lipschitz continuity in transferability rates for this paper. However, [1] offers a much more extensive theoretical exploration than Proposition 3.3 in this paper. Notably, [1] employs second-order Taylor expansion to conduct a fine-grained analysis of gradients, smoothness, and other related properties. In contrast, the proof in this paper simply leverages the Lipschitz constant to simplify the analysis.
> - **Insights on gradient magnitude**: The findings in [1] emphasize the critical role of gradient magnitude in transferability (see Theorem 2 in [1]), which is closely tied to the Lipschitz constant. In particular, **Lipschitz constant is the upper bound of gradient magnitude.** This connection suggests that deeper insights into gradient behavior could enrich the theoretical results in this paper. In other words, the analysis in Proposition 3.3 is less interesting than [1] since it only consider the upper bound of gradient magnitude. In contrast, [1] considers the gradient magnitude itself as well as the consine similarity, smoothness, etc.
> - **Transferability bounds**: The proof of Lemma 5 from [1] provides valuable insights into the upper bound of transferability. It demonstrates that transferability is constrained by the disagreement between two classifiers and the total variation distance between adversarial and clean data distributions (See $\operatorname{Pr}(f(\mathcal{A}(x)) \neq g(\mathcal{A}(x))) \leq \operatorname{Pr}(f(x) \neq g(x))+\rho$ in the proof of Lemma 5). Likewise, Proposition 3.3 in this paper bounds transferability using the product of the Lipschitz constant and radius, along with a disagreement term, i.e., $\left|R\_{\mathcal{F}}\left(\mathcal{A}\_{\mathcal{F}}(x), y\right)-R\_{\mathcal{G}}\left(\mathcal{A}\_{\mathcal{F}}(x), y\right)\right| \leq \left|R\_{\mathcal{F}}(x, y)-R\_{\mathcal{G}}(x, y)\right| + 2 L \epsilon$. I mean, they may bring similar insights. You may explain the connection between them.
>
> Given these points, I find that both the findings and proof techniques in Proposition 3.3 is covered by [1]. This overlap diminishes the novelty and theoretical significance of Proposition 3.3 in this paper. However, I still appreciate the authors' hard work and **I still like to keep my score as 6 if there is no other concern raised by other reviewers.**
>
> [1] TRS: Transferability Reduced Ensemble via Promoting Gradient Diversity and Model Smoothness. NeurIPS 2021.

---

> > ### Author Response · Authors · 2024-11-22
> >
> > We appreciate the reviewer for supporting our work while engaging in a constructive discussion and providing insightful comments. The responses to the reviewer’s concerns have been provided below.
> >
> > We have not claimed in our manuscript that proposition 3.3 is the main contribution of the paper; it is presented in the Motivation section of our manuscript to motivate the neglected adverse effect of the Lipschitz continuity on the transferability rate of adversarial examples, which we confirm **using empirical studies** (Figure 1). In fact, it is not even mentioned in the summary of our main contributions in section 1, where we specifically mention **We show the presence of a trade-off between single model robustness and ensemble robustness as the Lipschitz constant of the individual models changes through empirical analysis.** Therefore, we do not oversell the motivating hypothesis that proposition 3.3 provides as an insightful theoretical result, and that is indeed why we have called that a proposition, rather than a theorem.  Also, Lemma 5 of  TRS was one of the motivating results for our work. They showed more accurate models behave similarly on adversarial samples, which motivated us to draw a parallel for Lipschitz continuity of the models. We indeed have clearly mentioned the result of Lemma 5 of TRS in line 180 (Motivation section) of our manuscript.
> >
> >
> > We do believe our work is novel for the following reasons:
> >
> > - As far as we know, this is the first work that proposes orthogonalization of the corresponding affine layers of different models with respect to each other, and provides a practical solution for that. We clearly define how it can help with increasing the diversity of the models and provide a practical and effective method for that (see Figure 6 and Figure 7). The use cases of such orthogonalization can even go beyond ensemble robustness and might initiate other studies for other use cases in future work.
> >
> > - Pinpointing the adverse effect of Lipschitz continuity on the transferability rate through empirical analysis is a novel observation that can motivate further theoretical and empirical studies.
> >
> > - In terms of theoretical results, we believe our theorem 4.1 is indeed a novel finding which shows why our proposed method can be indeed effective and practical for convolutional layers. And therefore, unlike proposition 3.3, we have referred to our theorem 4.1 in the summary of our main contributions: “we theoretically and empirically show that our method is highly efficient for convolutional layers“.
> >
> > The TRS paper is an interesting work and we are familiar with their contributions. Their theoretical results rely on the assumption of smoothness (Lipschitz continuity of the gradients) which is a much more restrictive assumption that does not hold in practice; controlling that in the experiments is not easy and precise. Other terms appearing in the bound that correspond to supremum and infimum of the cosine similarity over the whole input space, while theoretically sound, will require best-effort heuristics in practice to approximate. On the other hand, the assumption that we make can be simply satisfied by just clipping the spectral norm of models such as ResNet18. This of course does not overshadow the insight that the theoretical results of TRS provides. But the assumptions that we have made in our paper is simple enough that allows us to test and evaluate our hypothesis empirically (Figure 1), which in turn motivates our novel approach for resolving that. Also, while theoretically interesting, TRS is very slow in practice and its true effectiveness in increasing robustness has been questioned by later works [1]. Our method shows increased robustness using different attacks, including MORA [1].
> >
> > As the reviewer suggested in their earlier comments, we are working on expanding the related works and motivation sections; this will further clarify the main contributions of this work in contrast to the existing ones. We thank the reviewer again for deadicating their time for improving the reviewing process.
> >
> > We do hope our responses have brought your attention to what we know as the main contributions of our work, and we hope the reviewer considers them in their evaluation of our work.
> > .
> >
> > [1]  Yu et al., MORA: Improving Ensemble Robustness Evaluation with Model-Reweighing Attack. NeurIPS 2022.

---

> > > ### Comment · Reviewer_HKeX · 2024-11-23
> > > **Thanks**
> > >
> > > I see, thank you very much for your patient response. It seems I misunderstood earlier, but I now understand your point. The main contribution of this paper leans more towards the algorithm and its corresponding theoretical analysis. While the importance of the Lipschitz constant has been hinted at in existing theories, this work is the first to emphasize its significance explicitly. Let’s wait for feedback from the other reviewers. Good luck!

---

### Official Review · Reviewer_Au68 · 2024-10-29

**Soundness:** 3
**Presentation:** 3
**Contribution:** 2
**Rating:** 6
**Confidence:** 3

**Summary:**

The paper analyzes the relationship between model Lipschitzness and the transferability of adversarial examples. Under the setting of TRS model ensemble, decreasing the model's Lipschitz constant improves standalone robustness but also increases transferability, which hurts ensemble robustness. To this end, the authors propose "LOTOS", which promotes the orthogonality of the singular vectors corresponding to the largest singular values of the transformation matrices of the layers of the ensemble candidate models. LOTOS can decrease transferability while only having a slight effect on individual robustness, thereby improving the ensemble robustness. Furthermore, LOTOS can work with heterogeneous model architectures in conjunction with adversarial training.

**Strengths:**

The paper analyzes an interesting problem, and uses the conclusions to propose a method that seems to work.

**Weaknesses:**

First of all, I would like to disclose that I am not an expert on black-box attacks or diversity-promoting robust ensembles. I am more familiar with the literature about the white-box scenario.

- The proposed method seems promising in the black-box setting, but how about the white-box case? I think this question is important because it will disentangle robustness from attack difficulty.
- What would happen if the "source model" in the black-box transfer attack is a LOTOS model itself (trained separately from the target model)?
- All definitions in this paper seem to build on the $\ell_2$ norm. Is the analyses and the proposed method only applicable to $\ell_2$ attacks? How about other attack budgets?
- Just to clarify, are all models in the ensemble trained simultaneously, or can they be trained sequentially? Asking for clarification here because the proposed loss function for each model depends on the weights of all other models.
- Line 285 says "for the orthogonalization to be effective in Equation (4), it is necessary to increase the value of $k$ (dimension of orthogonal sub-spaces)". Then, Line 294 says "$k = 1$ can be effective in orthogonalization with respect to the remaining singular vectors for convolutional layers." Are these not contradictory?
- In Theorem 4, the right-hand-side of Equation 5 has two terms under the square root. Which one is dominant? $\epsilon^2$ or the other terms?
- Line 333: typo "ensebmels".

**Questions:**

- The experiments in the paper focuses on ResNet and DLA. Do you think the method will extend to other popular architectures such as vision transformer, which operates differently and can be shallower?
- Line 215 (using the difference in the population loss of the two models on these adversarial examples as a proxy) -- How good is this approximation? If we use some alternative loss functions other than cross-entropy, is there any chance for this approximation become exact?
- In Equation (3), how would we determine $w_i$? Why do we need this weight, because intuitively we could perhaps just use the singular values here?
- Line 260 ($\\| \sum_{i=1}^d \sigma_i u_i v_i^\top v_i' \\|_2 = 0$). Here we only try to make the singular vectors orthogonal for each $i$, but the singular vectors for different $i$'s are not accounted. Would it possible to have the case where, for example, the first singular vector of $f^{(j)}$ is orthogonal to the first singular vector of $g^{(j)}$, but highly similar to the second singular vector of $g^{(j)}$? In this case, is the transferability between $f$ and $g$ still low?

---

> ### Author Response · Authors · 2024-11-17
>
> We thank the reviewer for their insightful comments. Below are our responses to their questions and concerns:
>
> ### **1.** How about the white-box case?
>
> We performed the same attack we used for the black-box setting, PGD-50 with $\ell_\infty \leq 0.04$, as well as two other stronger attacks AutoAttack [1] and MORA [2]:
>
>
> |               | PGD-50 | AutoAttack | MORA|
> | ----         | ----------- | ----------- | ------------ |
> | Orig       | $21.5 \pm 0.51$ | $18.89 \pm 0.48$ | $18.16 \pm 0.48$ |
> | C=1       | $37.4 \pm 0.37$ | $31.51 \pm 0.41$ | $30.98 \pm 0.51$ |
> | LOTOS | $39.9 \pm 0.71$ | $33.27 \pm 0.70$ | $32.41 \pm 0.43$ |
>
> However, we note that since our focus is on the transferability of adversarial examples from surrogate models and diversifying the models in the ensemble, our experiments were mainly on black-box attacks. For the black-box attacks, using Definition 3.2., we disentangle true transferability from the attack difficulty by conditioning on the examples that the attack is already successful on the surrogate model and computing the ratio of them that transfers to the target model. The same disentanglement is done from the accuracy of the models. This is in contrast to the previous definitions of transferability (e.g., [3]), in which using a stronger attack can lead to a higher transferability rate.
>
> Also, another motivation behind focusing on black-box attacks aligns with some recent efforts on training ensemble methods that are robust against transferability attacks because in most scenarios whitebox attacks are impractical [4].
>
>
>
> ### **2** Using LOTOS as the surrogate model?
>
> Here are the results that can be compared to the ensembles of three models in Table 2.
>
>  | Orig | C=1 | LOTOS |
>  | ----------- | ----------- | ------------ |
>  | $21.1 \pm 3.41$| $22.6 \pm 2.13$ | $24.7 \pm 2.91$ |
>
> As expected, using ensembles trained with LOTOS as the surrogate models leads to a stronger black-box attack against clipped models. Still, the ensembles trained with LOTOS are more robust against these attacks, as well.
>
> ### **3** All definitions in this paper seem to build on the $\ell_2$ norm.
>
> Robustifying the model by enforcing layer-wise Lipschitz constant has been utilized to guarantee $\ell_2$ robustness, which itself is an upper-bound for $\ell_\infty$ as well. Although LOTOS is proposed as a solution to this trade-off, by orthogonalizing the transformations, it promotes different behavior from the corresponding layers of the models with respect to any perturbation in the inputs. This leads to the diversity of the models regardless of the norm that has been used for computing the adversarial samples. In fact, in our experiments, we have used the default setting of the attacks from [5], which uses PGD-50 attack with $\ell_\infty$ norm.
>
> ### **4** Are all models in the ensemble trained simultaneously?
>
> The definition of the loss given in equation 4, as is, is applicable to simultaneous training of the models of the ensemble; however, we can also train them sequentially, and for training each new model make sure that its layers are orthogonal to the top-$k$ singular vectors of the corresponding layers of the previously trained models. We have not performed any experiment with the latter case, though, and as it is common in other robust ensemble training methods (TRS, DVERGE) only considered the case where the models are trained simultaneously. Still what you pointed out might be another benefit of using our method since it makes it easier to add additional models to the previously trained ensembles and reduces the required resources.
>
> ### **5** Line 285 says... Are these not contradictory?
>
> In line 285, we are discussing the general affine layers. In fact, in fully connected layers, we do need a high value of $k$ to ensure satisfactory orthogonalization because even if the top singular vector of a layer is orthogonal with respect to the other layer, still the other singular vectors can be aligned. Later in that section, we show that for “convolutional” layers, because of their intrinsic properties, even $k=1$ can be effective because it also induces orthogonality with respect to other singular vectors, to some extent (Theorem 4.1.). And that is what makes our approach highly efficient for convolutional layers. We will make this clear to avoid confusion.
>
>
> ### **6** In Theorem 4, which one is dominant?
>
> By using LOTOS, the value of $\epsilon$ can be forced to be very small (complete orthogonalization), but because of the limitations of convolutional layers and also minimizing the cross-entropy loss, it might not become zero for some layers. (see Figure 7 in Appendix). On the other hand, the second term can be forced to be smaller by choosing a higher weight decay. Also, it becomes naturally smaller for the more significant singular vectors by decreasing $\frac{p}{n}$. So the dominating factor depends on different choices of the hyper-parameters and we have not evaluated that.

---

> ### Author Response · Authors · 2024-11-17
>
> ### **7** Do you think the method will extend to other popular architectures such as vision transformers?
>
> Since prior works on ensemble robustness focused only on ResNet (or similar) models, we also chose them for our experiments. However, since our approach is applicable to any affine layer and is specifically much more effective for convolutional layers, it is interesting to see the effect of our approach on more complex models that contain these layers. That being said, that will be a stretch because for controlling the spectral norm, we have to train ensembles of these models from scratch and cannot use pre-trained models. Thus, it is unlikely that we will be able to provide these results in the brief rebuttal window.
>
> ### **8** Line 215  -- How good is this approximation?
>
> Proposition 3.3. is general and does not consider any specific loss function as long as the Lipschitz continuity condition holds. The point of this proposition is just to motivate that if we have the Lipschitz continuity property, the population loss on the adversarial samples would not differ much between the surrogate model and the target model, and that difference can depend on the Lipschitz constant. As you mentioned, the difference in the empirical loss would depend on the specific data samples and might vary to some extent, however, our empirical results show the same effect (e.g., Figure 1).
>
> ### **9** In Equation (3), how would we determine $w_i$?
>
> They are just hyper-parameters that can be tuned in hyper-parameter search. As the reviewer suggested, the singular values can be a reasonable and natural choice and that is what we considered as well. However, in our experiments, because we used $k=1$, that choice does not matter as we only have one singular vector and $w_1$ can be absorbed by $\lambda$ and be set to $1$.
>
> ### **10** Line 260...
>
> We should have replaced the index $i$ that goes from $1$ to $d$ in the summation to something else like $l$ to avoid confusion and make it like $|| \sum_{l=1}^d \sigma_l u_l v_l^\top v_i' ||_2 = 0$. Note that when $k=1$ we are orthogonalizing the largest singular vector of each layer with respect to the whole transformation of the other layer, which includes all its singular vectors. So the largest singular vector of each layer cannot be aligned with the second singular vector of the other layer or any of its other singular vectors.
> However, with $k=1$, we are only enforcing this orthogonalization for the first singular vectors of each layer with respect to the other layer, and still, the second singular vector of a layer can be aligned with the second singular vector of the other layer or any of its other singular vectors (except the first one). This in fact can be true for fully connected layers and is related to your other question and that is why for fully connected layers we need to consider higher values for $k$ for LOTOS to be effective. However, Theorem 4.1. proves that for convolutional layers by just orthogonalizing the first singular vectors, we get orthogonalization of the other singular vectors, to some extent, for free.
>
>
> [1] Croce et al., Reliable evaluation of adversarial robustness with an ensemble of diverse parameter-free attacks. ICML 2020.
>
> [2] Yu et al., MORA: Improving Ensemble Robustness Evaluation with Model-Reweighing Attack. NeurIPS 2022.
>
> [3] Yang, Z., Li, L., Xu, X., Zuo, S., Chen, Q., Zhou, P., ... & Li, B. (2021). Trs: Transferability reduced ensemble via promoting gradient diversity and model smoothness. Advances in Neural Information Processing Systems, 34, 17642-17655.
>
> [4] Sitawarin, C., Chang, J., Huang, D., Altoyan, W., & Wagner, D. PubDef: Defending Against Transfer Attacks From Public Models. In The Twelfth International Conference on Learning Representations.
>
> [5] https://github.com/AI-secure/Transferability-Reduced-Smooth-Ensemble

---

> > ### Author Response · Authors · 2024-11-27
> >
> > We thank the reviewer again for their constructive review leading to the improvement of our work!
> >
> > To summarize your review and our responses: In your initial review you mentioned that “The paper analyzes an interesting problem, and uses the conclusions to propose a method that seems to work”. However, you had some concerns regarding the clarification of the method and the lack of white-box attack experiments. We clarified our methodology and experiments and performed the requested experiments. We reported the results for multiple white-box attacks, including one attack specifically designed for model ensembles.
> >
> > We really hope that you are positive about our work and its findings and see this work as a useful addition to the community. If you believe that our proposed method is effective and the trade-off we have explored requires greater attention, we would appreciate it if you could consider raising your score to send a clear signal to the AC/SAC about your intentions in this competitive landscape.
> >
> > Please let us know if this resolves your concerns. We are happy to continue to engage!

---

### Official Review · Reviewer_uLeN · 2024-11-04

**Soundness:** 2
**Presentation:** 2
**Contribution:** 2
**Rating:** 5
**Confidence:** 4

**Summary:**

This paper finds that while Lipschitz continuity improves individual model robustness, it unexpectedly raises adversarial transferability in model ensembles. To counter this, the authors introduce LOTOS to boost ensemble robustness by promoting orthogonality between models.

**Strengths:**

This paper links decreasing Lipschitz continuity with increasing transferrability. This is an interesting point of exploration.

**Weaknesses:**

- What are the white-box attacks that you used in this paper? Some attacks may not be reliable in assessing ensemble robustness.
- The method appears to be an improvement upon TRS. However, TRS was shown to have an overestimated robustness [a], and it is not well-explained why the LOTOS loss is an improvement.
- It is not clear to me why DVERGE was not considered in the baseline comparisons, as it is known to be an effective robust baseline for ensemble defenses.
- Section 4 is poorly written. I am having trouble understanding your method. For eq. (3):
  - Where is $w_i$ introduced?
  - Why does this equation measure similarity?
  - What are $A$ and $B$ matrices, are they model parameters?
  - Also given that $A$ and $B$ are trainable, do you backprop through the SVD of them?
- For eq. (4): There is no $k$ in the LHS.
- Theorem 4.1:
  - "In Theorem 4.1, we prove even k = 1 can be effective in orthogonalization with respect to the remaining singular vectors for convolutional layers." It is not clear to me how it is proven for $k = 1$.
  - Why circular padding?
  - It is not clear to me how this theorem links with eq. (4).

**Questions:**

- Could you compare your methods with other ensemble defenses, especially with DVERGE?
- Could you consider stronger attacks such as AutoAttack [b] and MORA [a] which specifically target ensemble models?

[a]: Yu et al., MORA: Improving Ensemble Robustness Evaluation with Model-Reweighing Attack. NeurIPS 2022. https://arxiv.org/abs/2211.08008
[b]: Croce et al., Reliable evaluation of adversarial robustness with an ensemble of diverse parameter-free attacks. ICML 2020. https://arxiv.org/abs/2003.01690

---

> ### Author Response · Authors · 2024-11-17
>
> We thank the reviewer for their insightful comments. Below are our responses to their questions and concerns:
>
> ### **1** What are the white-box attacks that you used in this paper?
>
> As detailed in Appendix B.1, the white-box attack that we use is set up to evaluate the transferability rates among the models within an ensemble, and therefore it is not an attack on the whole ensemble. The attacks that we use for the whole ensemble are black-box attacks using surrogate models. The white-box attack, which is used only for measuring the transferability rate among the models within the ensemble, attacks each individual model using PGD-50 in a white-box manner to generate adversarial samples and then evaluates the other models of the ensemble on those samples to see what portion of them are transferred to them (according to Definition 3.2). We called it white-box to emphasize the use of one of the models within the ensemble as the surrogate model, but it is as if we are performing a black-box attack on the other models of the ensemble.
>
> ### **2** TRS was shown to have an overestimated robustness
>
> We would like to emphasize that LOTOS does not depend on TRS and the purpose of combining that with TRS is merely to show that it can be combined with other ensemble robustness approaches to further improve their robustness. LOTOS is a tangent to other defense mechanisms and has been proposed as a solution for increased transferability rate when Lipschitz continuity is used for the robustness of individual models.
>
> Lipschitz continuity has been used by prior works to make individual models more robust. Now assume that someone decides to use Lipschitz continuity to increase the robustness of each individual model in the ensemble and potentially incorporate that with another approach such as TRS to boost the robustness of the ensemble. We have shown in our work that in contrast to the expected outcome, the Lipschitz continuity that leads to more robust individual models also increases the transferability rate among them and prevents the desired boost in the robustness of the ensemble. LOTOS provides a solution to this increase in the transferability rate while allowing the ensemble to benefit from the robustness effect of Lipschitz continuity on individual models.
>
> Similarly, in Appendix B.8, we showed that LOTOS does not interfere with the robustness gains from other approaches designed for boosting the robustness of individual models, such as adversarial training. Our goal for the two experiments in 5.5 and B.8 is to show that LOTOS can be used alongside other defense mechanisms for adversarial robustness.
>
> ### **3** It is not clear to me why DVERGE was not considered in the baseline comparisons
>
> Thank you for bringing DVERGE to our attention. We will perform our experiments with DVERGE as well and present the results shortly.
>
> ### **4** For eq. (3):
>
> - Where is $w_i$ introduced?
>
> The line right after the equation explains that “$w_i$’s are arbitrary weights which are non-increasing with $i$ to emphasize “more importance” for the singular vectors corresponding to top singular values”
>
> - Why does this equation measure similarity?
>
> It measures how similar the transformations of two layers are to one another. The more aligned the singular vectors of the transformation matrices of affine layers are, the more similar they behave on the input space. Also, for each layer, the top singular vectors explain the majority of the transformation. Also note that as mentioned after the equation, the maximum value is attained when both layers are the same, and the minimum value (0) is attained when each layer is orthogonal with respect to the top $k$ singular vectors of the other layer. In general, the more aligned their top singular vectors are with one another, the higher the value will be.
>
> - What are $A$ and $B$ matrices, are they model parameters?
>
> As mentioned in the line before the equation they are the corresponding matrix transformations of two corresponding affine layers $f^{(j)}$ and $g^{(j)}$.
>
> - Also given that $A$ and $B$ are trainable, do you backprop through the SVD of them?
>
> When backpropagating through the loss in equation 4, the singular vectors are fixed. The backpropagation computes the gradient with respect to the parameters of layers $f$ and $g$ and updates them to minimize $S$. Then the new singular vectors of both layers are computed to recompute the Loss in equation 4 for the next iteration. Note that since the change in the parameters of the layers are small in each iteration, for computing the singular vectors, by reusing the singular vectors of the previous iteration during the PowerQR iteration, we converge to the correct singular vectors in a few steps of power iteration [4].

---

> ### Author Response · Authors · 2024-11-17
>
> ### **5** For eq. (4): There is no $k$ in the LHS.
>
> Equation 4 shows the training loss which constitutes the regular cross-entropy loss for training the models (left side of the ‘+’ sign) and LOTOS loss (right side of ‘+’ sign). $k$ is a hyper-parameter chosen apriori which affects the computation in equation 3 and defines the number of top singular vectors that we want to consider for orthogonalization.
>
> ### **6** In Theorem 4.1. it is not clear to me how it is proven for $k = 1$
>
> To make it more clear, first let's rewrite a simplified version of equation (3)  (without some of the hyper-parameters, such as $w_i$ and $mal$ for only one layer (so we can drop the $(j)$ superscript for clarity):
>
> $S_k(f,g) = \sum_{i=1}^k  ||f(v_i^\prime)||_2 + ||g(v_i)||_2 $.
>
> With the definitions in the theorem, we can simplify this further to:
>
> $\begin{align*} S_k(f,g) = \sum_{i=1}^k  || A v_i^\prime ||_2 + ||M_2(v_i)||_2 \end{align*}$
>
> Focusing on only the sum of the first term, the theorem proves that if we only enforce $||A v_1^\prime ||_2$ to be small, that also entails an upper-bound for $||A v_i^\prime ||_2, i \geq 2$, therefore, we do not need to use a large $k$ in the computation of equation 3 to ensure the orthogonalization; even using $k=1$ automatically implies small value for $||A v_i^\prime ||_2, i \geq 2$ and that makes it much more computationally efficient.
>
> ### **7** Why circular padding?
>
> The choice of circular padding for proving theoretical results on convolutional layers is a common practice ([1,2,3,4]) and allows for theoretical reasoning about these layers. Convolutional layers with other types of padding are expected to act similarly to the corresponding convolutional layers with circular padding ([1,3,5]). That being said, we emphasize that in our experiments we did not make any modification to the padding type (e.g., zero padding or no padding) or other configurations of the convolutional layers of the models that we used, which verifies the extension of our theoretical results to other types of padding as well.
>
> ### **8**  It is not clear to me how this theorem links with eq. (4).
>
> As mentioned in section 4, the higher values of $k$ entail a larger orthogonalization among the two corresponding affine layers, but increasing this value would make the computation of equation 4 more costly. This Theorem entails that, for convolutional layers, unlike dense layers, the value of $k$ does not need to be large for $S_k$ (defined in equation 3), and a small value of $k$ (e.g., 1) would imply the orthogonalization of the other singular vectors to some extent. On the contrary, for dense layers, this is not true; if one chooses $k=1$ for dense layers (e.g., fully connected layers), the only thing that LOTOS does is that it enforces the first singular vectors of the two layers to be orthogonal. However, the second singular vectors might be completely aligned with each other because they are independent of the directions of the first singular vectors.
>
> ### **8**  Could you consider stronger attacks such as AutoAttack [b] and MORA [a]?
>
> Here are the results on the robust accuracy against whitebox attacks on the whole ensemble (all the models of the ensemble are provided to the attacker):
>
> |               | PGD-50 | AutoAttack | MORA|
> | ----         | ----------- | ----------- | ------------ |
> | Orig       | $21.5 \pm 0.51$ | $18.89 \pm 0.48$ | $18.16 \pm 0.48$ |
> | C=1       | $37.4 \pm 0.37$ | $31.51 \pm 0.41$ | $30.98 \pm 0.51$ |
> | LOTOS | $39.9 \pm 0.71$ | $33.27 \pm 0.70$ | $32.41 \pm 0.43$ |
>
> The two new attacks are indeed more powerful in the whitebox setting. We also tried these two attacks in the black-box setting we used for our experiment, by attacking the surrogate ensemble and using the generated adversarial examples to evaluate the target ensemble:
>
> |               | PGD-50 | AutoAttack | MORA|
> | ----         | ----------- | ----------- | ------------ |
> | Orig       | $16.4 \pm 1.11$ | $16.3 \pm 0.69$| $30.0 \pm 0.52$ |
> | C=1       | $26.4 \pm 0.57$ | $30.0 \pm 0.39$ | $34.9 \pm 1.07$ |
> | LOTOS | $32.2 \pm 0.99$ | $33.7 \pm 0.76$ | $40.1 \pm 0.83$ |
>
> It is interesting to see that although these two attacks were stronger in the white-box setting, the adversarial examples they find might be too specific to the surrogate ensemble preventing them from generalizing to the target ensemble. Both results can be insightful and we will add them to our revised manuscript.

---

> ### Author Response · Authors · 2024-11-17
>
> ### **8**  Could you compare your methods with other ensemble defenses, especially with DVERGE
>
> We are running experiments for training DVERGE ensembles to compare the effect of incorporating LOTOS using our current attacks as well as AutoAttack and MORA to compare with the ones trained with TRS. We will report the results here once they are ready.
>
> [1] Sedghi, H., Gupta, V., & Long, P. M. (2018). The singular values of convolutional layers. arXiv preprint arXiv:1805.10408.
>
> [2] Senderovich, A., Bulatova, E., Obukhov, A., & Rakhuba, M. (2022). Towards practical control of singular values of convolutional layers. Advances in Neural Information Processing Systems, 35, 10918-10930.
>
> [3] Delattre, B., Barthélemy, Q., Araujo, A., & Allauzen, A. (2023, July). Efficient bound of Lipschitz constant for convolutional layers by gram iteration. In International Conference on Machine Learning (pp. 7513-7532). PMLR.
> [4] Boroojeny, A. E., Telgarsky, M., & Sundaram, H. (2024, April). Spectrum Extraction and Clipping for Implicitly Linear Layers. In International Conference on Artificial Intelligence and Statistics (pp. 2971-2979). PMLR.
>
> [5] Gray, R. M. (2006). Toeplitz and circulant matrices: A review. Foundations and Trends® in Communications and Information Theory, 2(3), 155-239.

---

> > ### Author Response · Authors · 2024-11-22
> >
> > We are done with training DVERGE ensembles and evaluating the effectiveness of LOTOS in improving their robustness. Below you can find the results on the robust accuracy against white-box attacks by MORA ($\epsilon=0.04$) on ensembles of ResNet18 models with batch norm layers (similar to the setting of section 5.5) trained with TRS and DVERGE. Indeed, as the reviewer mentioned, DVERGE ensembles seem to be more robust, and adding this result will further showcase the effectiveness of our method in improving the robustness of other methods when used together. As the results show the highest robust accuracy is achieved by combining DVERGE and LOTOS (D + LOTOS):
> >
> >
> >
> > |                     |   LOTOS   | TRS | TRS + C=1 | TRS + LOTOS | DVERGE | D + C=1  | D + LOTOS  |
> > | ----                | -------- | ----------- | ----------- | ------------ | ----------- | ----------- | ------------ |
> > | Robuat Acc  |  $16.8 \pm 0.51$        | $10.3 \pm 0.33$ | $13.2 \pm 1.12$| $14.5 \pm 0.81$ | $19.7 \pm 2.34$  | $26.8 \pm 0.75$ | $29.2 \pm 0.56$ |
> >
> > We thank the reviewer for pointing out this interesting prior work which further proves the effectiveness of our proposed method to be combined with prior ensemble robustness methods to improve their robustness. We believe we have provided responses to all the reviewer’s concerns and we hope the reviewer considers increasing their score toward acceptance.

---

> ### Comment · Reviewer_uLeN · 2024-11-24
>
> Thank you for your detailed response. I have raised my score as most of my confusion are addressed. It is promising that by combining DVERGE with LOTOS, it can achieve a much higher robust accuracy. However, the question regarding its competitiveness wrt DVERGE remains.

---

> ### Author Response · Authors · 2024-11-25
>
> We would like to thank the reviewer for going through our responses to their constructive comments and increasing their score. As the reviewer mentioned, using LOTOS with DVERGE ensembles increases their robust accuracy against MORA white-box attacks by $9.5$ percentage points which we believe is very notable and makes a great difference.
>
> DVERGE has proved to be an effective method for increasing ensemble robustness and it has introduced novel ideas for increasing model diversity through their distillation loss and cross-adversarial training. However, we believe **comparing LOTOS alone with DVERGE is not a fair comparison** because LOTOS increases the training time of ensembles of ResNet18 only by $2.4$ times per epoch (see Appendix B.5) whereas DVERGE increases the training time by a factor of $14.3$ because of all the different techniques they use in their algorithm. **If we only add adversarial training to LOTOS**, which leads to almost the same (a factor of 14.6) increase in the training time (see Appendix B.8) for a fair comparison, then the robust accuracy against MORA becomes $49.5%$ ($\pm 1.45$), which is **$29.8$ percentage point higher than DVERGE alone**. We reported results of LOTOS alone without its combination with adversarial training because we wanted to disentangle its effectiveness from other common techniques for boosting adversarial robustness. But we will add this result to the revised version of our manuscript to highlight the advantages of our proposed method.
>
> We want to emphasize that still LOTOS without adversarial training achieves high adversarial robustness, with much lower running time than other techniques such as TRS and DVERGE, and this is a very important factor because it makes LOTOS practical for larger models. So we believe all both DVERGE and LOTOS can have their place in the world and be used depending on the available resources and needs.
>
> With all of this additional experimentation we have conducted and the additional analysis we have provided on the run-time of different methods and our new result on robust accuracy of LOTOS with adversarial training against MORA attack, we hope the reviewer will consider increasing their score, particularly since all other reviewers have found our work an example of “empirical observation of interesting deep learning phenomena” and see LOTOS as “a solution that uniquely focuses on orthogonalizing affine layers, setting it apart from existing methods that target different aspects of model diversity”. Also, we believe the concerns raised by the reviewer have been experimentally resolved (including this particular note on fair comparisons between DVERGE and LOTOS) in this rebuttal window.

---

> > ### Author Response · Authors · 2024-12-02
> >
> > We thank the reviewer again for their constructive suggestions and for engaging in the discussion leading to the improvement of our work.
> >
> > To summarize our conversation: In your initial review you mentioned that “This paper links decreasing Lipschitz continuity with increasing transferability. This is an interesting point of exploration”. However, you had some concerns regarding the experiments with other SOTA methods and more recent attack methods. We performed all those experiments and reported the results which led you to raise your score. Still, you pointed out the shortcoming of our method in comparison to DVERGE, which we believe we justified by further results on the run-time comparison and reporting the results when adversarial training is added to LOTOS, which outperforms DVERGE by a large margin with a similar run-time.
> >
> > We really hope that you are positive about our work and its findings and see this work as a useful addition to the community which further motivates future research and explorations. If you believe that this phenomenon requires greater attention, and our findings and methodology are strong (which we showed through extensive experiments with your suggested methods), we humbly request you to consider raising your score because your score, we believe, is the main bottleneck. All the other reviewers have pointed out the merits of this work and reflected that in their scores. Here are some quotes from the other reviewers:
> >
> > - Reviewer oLSw: "I am a fan of establishing (through published literature) empirical observations of interesting deep learning phenomena. This is an example of such a thing." and "The evaluation is pretty strong and despite some obvious additions (that I mention later) I think it is sufficient to establish this as a phenomenon that might be real and that should be investigated further."
> >
> > - Reviewer Au68: "The paper analyzes an interesting problem, and uses the conclusions to propose a method that seems to work."
> >
> > - Reviewer HKeX: "It's interesting to observe the trade-off between the robustness of individual models and the transferability of adversarial examples when changing Lipschitz constant" and "The proposed LOTOS method is presented as a solution that uniquely focuses on orthogonalizing affine layers, setting it apart from existing methods that target different aspects of model diversity." Also, they mentioned: "The empirical results are shown clearly and shed light on so much understanding of the proposed solution"
> >
> > Please let us know if this resolves your concerns. We are happy to continue to engage.

---

### Official Review · Reviewer_oLSw · 2024-11-12

**Soundness:** 3
**Presentation:** 3
**Contribution:** 2
**Rating:** 6
**Confidence:** 3

**Summary:**

This paper makes 3 contributions to the field of robust ensemble training:

1. It identifies a trade-off in neural network ensembles: while decreasing the Lipschitz constant of individual models improves their robustness, it unfortunately also increases the transferability of adversarial examples between models in the ensemble, potentially undermining the ensemble's overall robustness.

2. It introduces LOTOS (Layer-wise Orthogonalization for Training rObust enSembles), a new training method that promotes orthogonality between corresponding layers of different models in an ensemble. LOTOS is computationally efficient and pretty architecturally agnostic.

3. Through experiments on CIFAR-10 and CIFAR-100, the authors demonstrate that LOTOS improves ensemble robustness against black-box attacks by 6 percentage points, and when combined with existing robust ensemble methods, enhances their performance by up to 10.7 percentage points.

**Strengths:**

Overall I'm a fan of establishing (through published literature) empirical observations of interesting deep learning phenomena. This is an example of such a thing. The fact that we can both a) make an individual model more robust via Lipschitz constant changes, but at the same time b) make an ensemble of such models more similar in a way that reduces the overall robustness of the ensemble, is a very curious piece of empirical evidence.

I think the evaluation is pretty strong and despite some obvious additions (that I mention later) I think it is sufficient to establish this as a phenomenon that might be real and that should be investigated further.

**Weaknesses:**

There are a few weakness that I believe could be relatively easily addressed.

### Limitation 1: Testing only against a single type of adversarial attack
As far as I understand it, the authors only test the robustness against one type of an adversarial attack -- the PGD-50 = Projected Gradient Descent with 50 iterations. I think it is important to 1) add more types of attacks (including black box ones), and 2) possibly look at some standard attack suites such as RobustBench (https://github.com/RobustBench/robustbench)

### Limitation 2: Adversarial robustness vs other kinds of robustness
What about other kinds of robustness such as robustness to noise, sheer, etc. Do you see similar phenomena to what you describe under adversarial attacks?

### Limitation 3: Attack strength norms and their effect on the phenomenon and LOTOS
I would be very curious what the effect of the attack norm (L_infty vs L_2 etc) and attack size (8/255, ...) is on the observed phenomenon and the efficacy of LOTOS. I can imagine that different attack size norms might lead to different qualitative tradeoffs. Could you please be more specific with what exactly you do and how does this relate to your theory & experiments?

### Limitation 4: Weak models?
The results and improvements you present are (I think) pretty non-robust to begin with. Am I wrong about this? How do robust model perform under the same attack? If the model you are using does not have a high robustness to begin with, I believe this might lead to misleading results on what helps and what doesn't. This is because many things tend to help when we're not close to the maximum robustness, but might not generalize. **Addressing this is crucial for me to raise your score.**

**Questions:**

I addressed the questions in the Weaknesses section.

---

> ### Author Response · Authors · 2024-11-17
>
> We thank the reviewer for their insightful comments. Below are our responses to their questions and concerns:
>
> ### **1.** Limitation 1:
>
> We performed additional experiments using AutoAttack [1] (used in evaluations of RobustBench) as well as MORA [2] (which is specifically designed for white-box attacks on ensembles and suggested by reviewer uLeN) to generate the adversarial examples on the surrogate models and used Definition 3.2 to compute the robust accuracy against transferability-based attacks (similar setting to the PGD-50 with $\ell_\infty \leq 0.04$ reported in Table 2).
>
> |               | PGD-50 | AutoAttack | MORA|
> | ----         | ----------- | ----------- | ------------ |
> | Orig       | $16.4 \pm 1.11$ | $16.3 \pm 0.69$| $30.0 \pm 0.52$ |
> | C=1       | $26.4 \pm 0.57$ | $30.0 \pm 0.39$ | $34.9 \pm 1.07$ |
> | LOTOS | $32.2 \pm 0.99$ | $33.7 \pm 0.76$ | $40.1 \pm 0.83$ |
>
>
> ### **2.** Limitation 2:
>
> In general, adversarial samples can be seen as the worst-case change within the used $\ell_p$ norm ball, and robustness against these samples entails accuracy against other changes within that $\ell_p$ norm ball. Still, to evaluate these methods on other types of noise, we used the benchmark introduced by [3] for CIFAR-10 and report the results in the table below:
>
> | | Gaussian noise | Gaussian blur| Impulse noise| Motion blur| pixelate| fog| frost| saturate| Shot noise| Zoom blur| spatter| Speckle noise| **AVG** |
> | ----         | ----------- | ----------- | ------------ | ---- | ---- | ---- | ---- | ---- | ---- |---- |---- |---- |---- |
> | Orig       | 36.1 | 55.7| 27.0 | 71.2 | 44.3 |  **77.5** | 62.5 |  **87.4** | 42.9 | 71.8 | 73.5 | 45.8 | 58.0 |
> | C=1       | 56.0 | 63.7 | 37.8 | 71.6 | 69.0 | 64.9 |72.8 | 84.6 | 60.1 |  **74.1** | 77.6 | 58.9 | 65.9 |
> | LOTOS |  **61.6** |  **63.8** | **44.5** | **71.8**|  **70.8** | 63.3 |  **74.8** | 84.4 |  **64.5** | 73.8 |  **78.2** | **62.7** | **67.9** |
>
> As the results show C=1 ensembles work better than the original ensembles because they are trained to be less sensitive to changes in the inputs. LOTOS ensembles are more robust than the other two because, in addition to benefiting from the Lipschitz continuity introduced in C=1 ensembles, they further diversify the models for how to react to changes in the input.
>
>
> ### **3.** Limitation 3:
>
> The enforced Lipschitz continuity on the models of the ensemble provides robustness against a certain $\ell_2$ norm radius of the samples. We have used a surrogate ensemble and used PGD-50 with $\ell_\infty \leq 0.04$ to generate adversarial examples. The choice of $\ell_\infty$ allows the adversarial attack to be more flexible and therefore more difficult to defend against [4]. We then measure the transferability rate using Definition 3.2. Here, we have performed some experiments with other attack sizes, as well as other norms ($\ell_2$ and $\ell_1$). As we can see with larger attack sizes the effectiveness of our method becomes more clear.
>
> Results for PGD-50 with different attack sizes:
>
> |               | $\epsilon=0.02$ | $\epsilon=0.04$ | $\epsilon=0.08$ |
> | ----         | ----------- | ----------- | ------------ |
> | Orig       | $22.9 \pm 1.04$ | $16.4 \pm 1.11$ | $8.2 \pm 0.63$ |
> | C=1       | $28.2 \pm 1.03$ | $26.4 \pm 0.57$ | $20.7 \pm 0.50$ |
> | LOTOS | $33.1 \pm 1.14$ | $32.2 \pm 0.99$ | $28.1 \pm 1.63$ |
>
> Results for PGD-50 with $\ell_1 \leq 0.04$ and  $\ell_2 \leq 0.04$:
>
> |               | $\ell_\infty$ | $\ell_2$ | $\ell_1$ |
> | ----         | ----------- | ----------- | ------------ |
> | Orig       | $16.4 \pm 1.11$ | $20.4 \pm 1.00$ | $20.2 \pm 0.98$ |
> | C=1       | $26.4 \pm 0.57$ | $17.9 \pm 1.20$ | $17.9 \pm 1.12$ |
> | LOTOS | $32.2 \pm 0.99$ | $22.2 \pm 1.79$ | $22.1 \pm 2.14$ |

---

> ### Author Response · Authors · 2024-11-17
>
> ### **4.** Limitation 4: Weak models?
>
> As the reviewer mentioned, the models that have a lot of room for improvement from the robustness perspective are easier to modify to increase their robustness. Although we used ResNet and DLA models in our experiments (which is common practice for prior works on ensemble robustness), we tried to show LOTOS can improve the robustness, whether it is used for non-robust models (Table 1), or when used with models that are made individually robust using adversarial training (Table 6 in Appendix B.8), or when used with ensembles that are made robust using other ensemble robustness methods (Table 3). All these models have different robustness levels to begin with, but we showed that LOTOS consistently makes an improvement in the robustness of these models. Note that in addition to these experiments, we also performed some experiments on ResNet models without batch norm layers, which has been shown to improve the robustness of the models (Table 5).
>
> Furthermore, we have shown that LOTOS improves over models that are Lipschitz continuous, which are shown to be much more robust and target reaching the maximum robustness by limiting the change in the output for small changes in the input [5,6,7]. However, this approach has the robustness of individual models as the goal and what we have shown is that they will be less effective when used in the form of an ensemble.
>
> To further address the reviewer’s concerns, we believe the rebuttal timeline would allow us to perform some of our basic experiments with ResNet50 models as well, which have been shown to be much more robust than ResNet18 models [8]. We will report the results here once they are ready. Also note that we already have some experiments that include ResNet34 models, which are more robust than ResNet18 models as well (see Section 5.4 and Appendix B.7).
>
> [1] Croce et al., Reliable evaluation of adversarial robustness with an ensemble of diverse parameter-free attacks. ICML 2020.
>
> [2] Yu et al., MORA: Improving Ensemble Robustness Evaluation with Model-Reweighing Attack. NeurIPS 2022.
>
> [3] Hendrycks, D., & Dietterich, T. (2018, September). Benchmarking Neural Network Robustness to Common Corruptions and Perturbations. In International Conference on Learning Representations.
>
> [4] Yang, G., Duan, T., Hu, J. E., Salman, H., Razenshteyn, I., & Li, J. (2020, November). Randomized smoothing of all shapes and sizes. In International Conference on Machine Learning (pp. 10693-10705). PMLR.
>
> [5] Szegedy, C. (2013). Intriguing properties of neural networks. arXiv preprint arXiv:1312.6199.
>
> [6] Cohen, J., Rosenfeld, E., & Kolter, Z. (2019, May). Certified adversarial robustness via randomized smoothing. In international conference on machine learning (pp. 1310-1320). PMLR.
>
> [7] Boroojeny, A. E., Telgarsky, M., & Sundaram, H. (2024, April). Spectrum Extraction and Clipping for Implicitly Linear Layers. In International Conference on Artificial Intelligence and Statistics (pp. 2971-2979). PMLR.
>
> [8] Peng, S., Xu, W., Cornelius, C., Li, K., Duggal, R., Chau, D. H., & Martin, J. (2023). Robarch: Designing robust architectures against adversarial attacks. arXiv preprint arXiv:2301.03110.

---

> > ### Author Response · Authors · 2024-11-20
> >
> > We are done with the ResNet50 experiments regarding Limitation 4. We performed the same type of black-box attack to evaluate the effectiveness of LOTOS in robustness against transferability attacks and have shown the results next to what we had for ResNet18 ensembles:
> >
> > |               | ResNet18 | ResNet50 |
> > | ----         | ----------- | ----------- |
> > | Orig       |  $16.4 \pm 1.11$  | $27.0 \pm 1.03$ | l
> > | C=1       |  $26.4 \pm 0.57$  | $23.7 \pm 1.29$ |
> > | LOTOS  |  $32.2 \pm 0.99$  | $34.8 \pm 1.59$ |
> >
> > As the results show the Orig ensembles are more robust, but the Lipschitz continuity (C=1) leads to further decrement of robustness while LOTOS still improves the robustness compared to the ResNet18 ensemble.
> >
> > Please let us know if this experiment along with our earlier response addresses the reviewer's concern. Thanks!

---

> > > ### Author Response · Authors · 2024-11-25
> > >
> > > We would like to inform the reviewer of the new results that we prepared in response to reviewer uLeN because we think it is also relevant to Limitation 4 on "non-robust models to begin with". We performed a white-box attack using MORA ($\epsilon=0.04$) on ensembles of ResNet-18 models with batch norm layers (similar to the setting of section 5.5) trained with TRS and DVERGE. DVERGE ensembles seem to be more robust than TRS, and both are more robust than regular ensembles (so they have **different levels of robustness to begin with**). But, as the results show combining LOTOS with any of these methods, which are at different levels of robustness, improves their robust accuracy. It improves the robustness of TRS and DVERGE ensembles by 4.2 and 9.5 percentage points.
> > >
> > >
> > > |                     |   LOTOS   | TRS | TRS + C=1 | TRS + LOTOS | DVERGE | D + C=1  | D + LOTOS  |
> > > | ----                | -------- | ----------- | ----------- | ------------ | ----------- | ----------- | ------------ |
> > > | Robuat Acc  |  $16.8 \pm 0.51$        | $10.3 \pm 0.33$ | $13.2 \pm 1.12$| $14.5 \pm 0.81$ | $19.7 \pm 2.34$  | $26.8 \pm 0.75$ | $29.2 \pm 0.56$ |
> > >
> > > Note that LOTOS alone is less robust than DVERGE alone, but that does not contradict our claim about our method regarding its ability to enhance the robustness of prior methods. Also, as mentioned to reviewer uLeN, it is not a fair comparison because LOTOS increases the training time of ensembles of ResNet18 only by $2.4$ times per epoch (see Appendix B.5) whereas DVERGE increases the training time by a factor of $14.3$ because of all the different techniques (e.g., distillation and cross-adversarial training) they use in their algorithm. **If we only add adversarial training to LOTOS**, which leads to almost the same (a factor of 14.6) increase in the training time (see Appendix B.8) for a fair comparison, then the robust accuracy against MORA becomes $49.5%$ ($\pm 1.45$), which is **$29.8$ percentage point higher than DVERGE alone**.
> > >
> > >
> > > Because of encouraging comments from the reviewer about our work and since we believe we have addressed their concerns through various experiments, we would appreciate it if the reviewer could consider increasing their score to send a clear signal to the AC.

---

> > > > ### Author Response · Authors · 2024-11-27
> > > >
> > > > We thank the reviewer again for their constructive suggestions leading to the improvement of our work.
> > > >
> > > > To summarize your initial review: In your very encouraging review, you mentioned that you are “a fan of establishing (through published literature) empirical observations of interesting deep learning phenomena. This is an example of such a thing.” Also, you mentioned, “the evaluation is pretty strong and despite some obvious additions (that I mention later) I think it is sufficient to establish this as a phenomenon that might be real and that should be investigated further”. In addition, you mentioned addressing your concerns might lead to raising your score.
> > > >
> > > > You suggested some additional experiments (e.g., other attack methods, different attack sizes, other norm bounds, non-adversarial noise, and models with different levels of robustness) to further improve our work. We performed all those experiments and reported the results which further showed the consistent benefits of our proposed method. We really hope that you are still positive about our work and its findings; the revised version reflects all your suggestions. Should you believe that this phenomenon requires greater attention and that our findings and methodology are strong (as you mentioned in your initial review), we would appreciate it if you could consider raising your score to send a clear signal to the AC/SAC about your intentions in this competitive landscape.
> > > >
> > > > Please let us know if this resolves your concerns. We are happy to continue to engage.

---

### Comment · Area_Chair_JE6T · 2024-11-24
**Reminder - Public Discussion Phase Ending Soon**

Dear PC memebers,

Thank you for your valuable comments during the review period, which raised many interesting and insightful questions. I am also glad to see that we have lots of discussions for this paper. Now the discussion period is coming to a close, please take a moment to review the authors’ responses if you haven’t done so already. Even if you decide not to update your evaluation, kindly confirm that you have reviewed the responses and that they do not change your assessment.

Timeline: As a reminder, the review timeline is as follows:

November 26: Last day for reviewers to ask questions to authors.
November 27: Last day for authors to respond to reviewers.
November 28 - December 10: Reviewer and area chair discussion phase.
December 20: Meta reviews and initial decisions are due.

Thank you for your time and effort!

Best regards,
AC

---

### Author Response · Authors · 2024-11-26
**Rebuttal Revision**

We thank all the reviewers for providing insightful comments and all the encouragement we received for our work. We believe addressing these concerns has made our work stronger and further showcases its effectiveness. The reviewers' comments focused on making connections to prior work, clarifying our proposed method and discussions, using more ensemble robustness methods in our experiments, and exploring other attack algorithms, including white-box attacks specifically designed for evaluating ensemble robustness. We addressed these concerns by:

- Making detailed clarification on the parts of our manuscript that raised confusion for the reviewers. In particular, we further clarified the explanation of our methods and the experiments of our work (see line 259, line 856, and line 870 of the revised version).

- Making connections to other theoretical results by prior work and explaining how the trade-off observed by us fits the theoretical framework in these works. In particular, we show the recent theoretical results in [1] regarding the trade-off of complexity and diversity of the ensemble models for transferability attacks aligns with our finding about the trade-off caused by Lipschitz continuity as this leads to less complex models which in turn lowers the diversity needed for defense against transferability attacks (see Appendix C.1 in the revised manuscript for more details).

- Adding additional experiments on other attack methods, attack sizes, and perturbation norms. Our results show the effectiveness of our proposed method in all different cases and the effectiveness becomes even clearer as the attack becomes stronger (see Appendix B.10.1 and Appendix B.10.2 in the revised version).

- Evaluating our methods against benchmarks [2] for other types of noise (e.g., Gaussian noise, spatter, etc.). Our results show that LOTOS leads to more robustness against most of the noise types and leads to the highest robust accuracy on average (see Appendix B.11 of the revised version).

- Adding the results for white-box attacks including the ones suggested by the reviewers (AutoAttack [3] and MORA [4]) to evaluate the effectiveness of our methods. These attacks are stronger attacks and the higher robust accuracy of LOTOS further showcases its effectiveness (see Appendix B.10.4 of the revised version).

- Adding new models (ResNet50 that are shown to be more robust [5]) to showcase the generality of our proposed method for models with different levels of robustness (see Appendix B.10.3 of the revised version).

- Adding a new SOTA ensemble robustness method (DVERGE [6]) to evaluate the effectiveness of LOTOS in improving its robustness. These ensembles are much stronger than TRS against MORA attacks, but the results show that adding LOTOS can further improve their robustness by 9.5 percentage points (see appendix B.9 of the revised version).

We believe we have addressed all the concerns raised by the reviewers in their individual responses and have incorporated the updates into the revised version of our manuscript before the discussion deadline. We would appreciate it if they could review their responses to identify any remaining concerns.

[1] Yao W, Zhang Z, Tang H, et al. Understanding Model Ensemble in Transferable Adversarial Attack[J]. arXiv preprint arXiv:2410.06851, 2024.

[2] Hendrycks, D., & Dietterich, T. (2018, September). Benchmarking Neural Network Robustness to Common Corruptions and Perturbations. In International Conference on Learning Representations.

[3] Croce et al., Reliable evaluation of adversarial robustness with an ensemble of diverse parameter-free attacks. ICML 2020.

[4] Yu et al., MORA: Improving Ensemble Robustness Evaluation with Model-Reweighing Attack. NeurIPS 2022.

[5] Peng, S., Xu, W., Cornelius, C., Li, K., Duggal, R., Chau, D. H., & Martin, J. (2023). Robarch: Designing robust architectures against adversarial attacks. arXiv preprint arXiv:2301.03110.

[6] Yang, Huanrui, et al. "Dverge: diversifying vulnerabilities for enhanced robust generation of ensembles." Advances in Neural Information Processing Systems 33 (2020): 5505-5515.

---

### Meta-Review · Area_Chair_JE6T · 2024-12-19

**Metareview:**

This paper discusses robust ensemble training based on the observation that "decreasing Lipschitz continuity corresponds to increasing transferability." While the idea is not particularly surprising, and the analysis lacks novelty (a sentiment shared by Reviewer HKek and myself), one reviewer noted that this is the first time its significance has been explicitly emphasized. Moreover, the authors proposed a new method, "LOTOTS," to promote orthogonality, which has been recognized by the reviewers as a notable technical contribution. Despite some weaknesses in the theoretical analysis, the method shows promise and adds value to the field.

Overall, I believe this is a borderline paper, but I am inclined to recommend acceptance.

**Additional Comments On Reviewer Discussion:**

The discussion on this paper has been generally sufficient. Three reviewers agreed that the paper highlights an interesting phenomenon, while Reviewer HKeX argued that similar ideas have already been analyzed. After the rebuttal, the consensus was that although this phenomenon has been previously touched upon, it has not been explicitly discussed, nor has any algorithm been designed based on it.

Other concerns were primarily related to the experiments and were largely addressed during the rebuttal.

The final scores reflect a weak accept, indicating that while the paper does not have significant weaknesses, it also does not present any groundbreaking advancements.

---

### Decision · Program_Chairs · 2025-01-22

Accept (Poster)